# StyleTTS 2: Towards Human-Level Text-to-Speech through Style Diffusion and Adversarial Training with Large Speech Language Models

**Yinghao Aaron Li    Cong Han    Vinay S. Raghavan**
**Gavin Mischler    Nima Mesgarani**
Columbia University
{yl4579,ch3212,vsr2119,gm2944,nm2764}@columbia.edu

## Abstract

In this paper, we present StyleTTS 2, a text-to-speech (TTS) model that leverages style diffusion and adversarial training with large speech language models (SLMs) to achieve human-level TTS synthesis. StyleTTS 2 differs from its predecessor by modeling styles as a latent random variable through diffusion models to generate the most suitable style for the text without requiring reference speech, achieving efficient latent diffusion while benefiting from the diverse speech synthesis offered by diffusion models. Furthermore, we employ large pre-trained SLMs, such as WavLM, as discriminators with our novel differentiable duration modeling for end-to-end training, resulting in improved speech naturalness. StyleTTS 2 surpasses human recordings on the single-speaker LJSpeech dataset and matches it on the multispeaker VCTK dataset as judged by native English speakers. Moreover, when trained on the LibriTTS dataset, our model outperforms previous publicly available models for zero-shot speaker adaptation. This work achieves the first human-level TTS on both single and multispeaker datasets, showcasing the potential of style diffusion and adversarial training with large SLMs. The audio demos and source code are available at `https://styletts2.github.io/`.

## 1   Introduction

Text-to-speech (TTS) synthesis has seen significant advancements in recent years, with numerous applications such as virtual assistants, audiobooks, and voice-over narration benefiting from increasingly natural and expressive synthetic speech [1, 2]. Some previous works have made significant progress towards human-level performance [3, 4, 5]. However, the quest for robust and accessible human-level TTS synthesis remains an ongoing challenge because there is still room for improvement in terms of diverse and expressive speech [5, 6], robustness for out-of-distribution (OOD) texts [7], and the requirements of massive datasets for high-performing zero-shot TTS systems [8].

In this paper, we introduce StyleTTS 2, an innovative TTS model that builds upon the style-based generative model StyleTTS [6] to present the next step towards human-level TTS systems. We model speech styles as a latent random variable and sample them with a probabilistic diffusion model, allowing the model to efficiently synthesize highly realistic speech without the need for reference audio. Since it only needs to sample a style vector instead of the entire speech as a latent variable, StyleTTS 2 is faster than other diffusion TTS models while still benefiting from the diverse speech synthesis enabled by diffusion models. One of the key contributions of StyleTTS 2 is the use of large pre-trained speech language models (SLMs) like Wav2Vec 2.0 [9], HuBERT [10], and WavLM [11] as discriminators, in conjunction with a novel differentiable duration modeling approach. This end-to-end (E2E) training setup leverages SLM representations to enhance the naturalness of the synthesized speech, transferring knowledge from large SLMs for speech generation tasks.

37th Conference on Neural Information Processing Systems (NeurIPS 2023).

Our evaluations suggest that speech generated by StyleTTS 2 surpasses human recordings as judged by native English speakers on the benchmark LJSpeech [12] dataset with statistically significant comparative mean opinion scores (CMOS) of $+0.28$ ($p < 0.05$). Additionally, StyleTTS 2 advances the state-of-the-art by achieving a CMOS of $+1.07$ ($p \ll 0.01$) compared to NaturalSpeech [5]. Furthermore, it attains human-level performance on the multispeaker VCTK dataset [13] in terms of naturalness (CMOS = $-0.02$, $p \gg 0.05$) and similarity (CMOS = $+0.30$, $p < 0.1$) to the reference speaker. When trained on a large number of speakers like the LibriTTS dataset [14], StyleTTS 2 demonstrates potential for speaker adaptation. It surpasses previous publicly available models in this task and outperforms Vall-E [8] in naturalness. Moreover, it achieves slightly worse similarity to the target speaker with only a 3-second reference speech, despite using around 250 times less data compared to Vall-E, making it a data-efficient alternative for large pre-training in the zero-shot speaker adaptation task. As the first model to achieve human-level performance on publicly available single and multispeaker datasets, StyleTTS 2 sets a new benchmark for TTS synthesis, highlighting the potential of style diffusion and adversarial training with SLMs for human-level TTS synthesis.

## 2 Related Work

**Diffusion Models for Speech Synthesis.** Diffusion models have gained traction in speech synthesis due to their potential for diverse speech sampling and fine-grained speech control [15]. They have been applied to mel-based text-to-speech [16, 17, 18, 19, 20], mel-to-waveform vocoder [21, 22, 23, 24, 25, 26], and end-to-end speech generation [27, 28, 29]. However, their efficiency is limited compared to non-iterative methods, like GAN-based models [30, 31, 32], due to the need to iteratively sample mel-spectrograms, waveforms, or other latent representations proportional to the target speech duration [15]. Furthermore, recent works suggest that state-of-the-art GAN-based models still perform better than diffusion models in speech synthesis [26, 33]. To address these limitations, we introduce style diffusion, where a fixed-length style vector is sampled by a diffusion model conditioned on the input text. This approach significantly improves model speed and enables end-to-end training. Notably, StyleTTS 2 synthesizes speech using GAN-based models, with only the style vector dictating the diversity of speech sampled. This unique combination allows StyleTTS 2 to achieve high-quality synthesis with fast inference speed while maintaining the benefits of diverse speech generation, further advancing the capabilities of diffusion models in speech synthesis.

**Text-to-Speech with Large Speech Language Models.** Recent advancements have proven the effectiveness of large-scale self-supervised speech language models (SLMs) in enhancing text-to-speech (TTS) quality [34, 35, 36, 37] and speaker adaptation [8, 38, 29, 39]. These works typically convert text input into either continuous or quantized representations derived from pre-trained SLMs for speech reconstruction. However, SLM features are not directly optimized for speech synthesis, while tuning SLMs as a neural codec [34, 35, 8, 29] involves two-stage training. In contrast, our model benefits from the knowledge of large SLMs via adversarial training using SLM features without latent space mapping, thus directly learning a latent space optimized for speech synthesis like other end-to-end (E2E) TTS models. This innovative approach signifies a new direction in TTS with SLMs.

**Human-Level Text-to-Speech.** Several recent works have advanced towards human-level TTS [3, 4, 5] using techniques like BERT pre-training [4, 40, 7] and E2E training [32, 5] with differentiable duration modeling [41, 42]. VITS [3] demonstrates MOS comparable to human recordings on the LJSpeech and VCTK datasets, while PnG-BERT [4] obtains human-level results on a proprietary dataset. NaturalSpeech [5], in particular, achieves both MOS and CMOS on LJSpeech statistically indistinguishable from human recordings. However, we find that there is still room for improvement in speech quality beyond these state-of-the-art models, as we attain higher performance and set a new standard for human-level TTS synthesis. Furthermore, recent work shows the necessity for disclosing the details of evaluation procedures for TTS research [43]. Our evaluation procedures are detailed in Appendix E, which can be used for reproducible future research toward human-level TTS.

## 3 Methods

### 3.1 StyleTTS Overview

StyleTTS [6] is a non-autoregressive TTS framework using a style encoder to derive a style vector from reference audio, enabling natural and expressive speech generation. The style vector is incorpo-

rated into the decoder and duration and prosody predictors using adaptive instance normalization (AdaIN) [44], allowing the model to generate speech with varying duration, prosody, and emotions.

The model comprises eight modules, organized into three categories: (1) a speech generation system (acoustic modules) with a text encoder, style encoder, and speech decoder; (2) a TTS prediction system with duration and prosody predictors; and (3) a utility system for training, including a discriminator, text aligner, and pitch extractor. It undergoes a two-stage training process: the first stage trains the acoustic modules for mel-spectrogram reconstruction, and the second trains TTS prediction modules using the fixed acoustic modules trained in the first stage.

In the first stage, the text encoder $T$ encodes input phonemes $\boldsymbol{t}$ into phoneme representations $\boldsymbol{h}_{\text{text}} = T(\boldsymbol{t})$, while the text aligner $A$ extracts speech-phoneme alignment $\boldsymbol{a}_{\text{algn}} = A(\boldsymbol{x}, \boldsymbol{t})$ from input speech $\mathbf{x}$ and phonemes $\boldsymbol{t}$ to produce aligned phoneme representations $\boldsymbol{h}_{\text{algn}} = \boldsymbol{h}_{\text{text}} \cdot \boldsymbol{a}_{\text{algn}}$ via dot product. Concurrently, the style encoder $E$ obtains the style vector $\boldsymbol{s} = E(\mathbf{x})$, and the pitch extractor $F$ extracts the pitch curve $p_{\boldsymbol{x}} = F(\boldsymbol{x})$ along with its energy $n_{\boldsymbol{x}} = \|\boldsymbol{x}\|$. Lastly, the speech decoder $G$ reconstructs $\hat{\boldsymbol{x}} = G\left(\boldsymbol{h}_{\text{algn}}, \boldsymbol{s}, p_{\boldsymbol{x}}, n_{\boldsymbol{x}}\right)$, which is trained to match input $\boldsymbol{x}$ using a $L_1$ reconstruction loss $\mathcal{L}_{\text{mel}}$ and adversarial objectives $\mathcal{L}_{\text{adv}}, \mathcal{L}_{\text{fm}}$ with a discriminator $D$. Transferable monotonic aligner (TMA) objectives are also applied to learn optimal alignments (see Appendix G for details).

In the second stage, all components except the discriminator $D$ are fixed, with only the duration and prosody predictors being trained. The duration predictor $S$ predicts the phoneme duration with $\boldsymbol{d}_{\text{pred}} = S(\boldsymbol{h}_{\text{text}}, \boldsymbol{s})$, whereas the prosody predictor $P$ predicts pitch and energy as $\hat{p}_{\boldsymbol{x}}, \hat{n}_{\boldsymbol{x}} = P(\boldsymbol{h}_{\text{text}}, \boldsymbol{s})$. The predicted duration is trained to match the ground truth duration $\boldsymbol{d}_{\text{gt}}$ derived from the summed monotonic version of the alignment $\boldsymbol{a}_{\text{algn}}$ along the time axis with an $L_1$ loss $\mathcal{L}_{\text{dur}}$. The predicted pitch $\hat{p}_{\boldsymbol{x}}$ and energy $\hat{n}_{\boldsymbol{x}}$ are trained to match the ground truth pitch $p_{\boldsymbol{x}}$ and energy $n_{\boldsymbol{x}}$ derived from pitch extractor $F$ with $L_1$ loss $\mathcal{L}_{f0}$ and $\mathcal{L}_n$. During inference, $\boldsymbol{d}_{\text{pred}}$ is used to upsample $\boldsymbol{h}_{\text{text}}$ through $\boldsymbol{a}_{\text{pred}}$, the predicted alignment, obtained by repeating the value 1 for $\boldsymbol{d}_{\text{pred}}[i]$ times at $\ell_{i-1}$, where $\ell_i$ is the end position of the $i^{\text{th}}$ phoneme $\boldsymbol{t}_i$ calculated by summing $\boldsymbol{d}_{\text{pred}}[k]$ for $k \in \{1, \ldots, i\}$, and $\boldsymbol{d}_{\text{pred}}[i]$ are the predicted duration of $\boldsymbol{t}_i$. The mel-spectrogram is synthesized by $\boldsymbol{x}_{\text{pred}} = G(\boldsymbol{h}_{\text{text}} \cdot \boldsymbol{a}_{\text{pred}}, E(\tilde{\boldsymbol{x}}), \hat{p}_{\tilde{\boldsymbol{x}}}, \hat{n}_{\tilde{\boldsymbol{x}}})$ with $\tilde{\boldsymbol{x}}$ an arbitrary reference audio that influences the style of $\boldsymbol{x}_{\text{pred}}$, which is then converted into a waveform using a pre-trained vocoder.

Despite its state-of-the-art performance in synthesizing diverse and controllable speech, StyleTTS has several drawbacks, such as a two-stage training process with an additional vocoding stage that degrades sample quality, limited expressiveness due to deterministic generation, and reliance on reference speech hindering real-time applications.

### 3.2 StyleTTS 2

StyleTTS 2 improves upon the StyleTTS framework, resulting in a more expressive text-to-speech (TTS) synthesis model with human-level quality and improved out-of-distribution performance. We introduce an end-to-end (E2E) training process that jointly optimizes all components, along with direct waveform synthesis and adversarial training with large speech language models (SLMs) enabled by our innovative differentiable duration modeling. The speech style is modeled as a latent variable sampled through diffusion models, allowing diverse speech generation without reference audio. We outline these important changes in the following sections with an overview in Figure 1.

### 3.2.1 End-to-End Training

E2E training optimizes all TTS system components for inference without relying on any fixed components like pre-trained vocoders that convert mel-spectrograms into waveforms [3, 32]. To achieve this, we modify the decoder $G$ to directly generate the waveform from the style vector, aligned phoneme representations, and pitch and energy curves. We remove the last projection layer for mel-spectrograms of the decoder and append a waveform decoder after it. We propose two types of decoders: HifiGAN-based and iSTFTNet-based. The first is based on Hifi-GAN [30], which directly generates the waveform. In contrast, the iSTFTNet-based decoder [45] produces magnitude and phase, which are converted into waveforms using inverse short-time Fourier transform for faster training and inference. We employ the snake activation function [46], proven effective for waveform generation in [31]. An AdaIN module [44] is added after each activation function to model the style dependence of the speech, similar to the original StyleTTS decoder. We replace the mel-discriminator in [6] with the multi-period discriminator (MPD) [30] and multi-resolution discriminator (MRD)

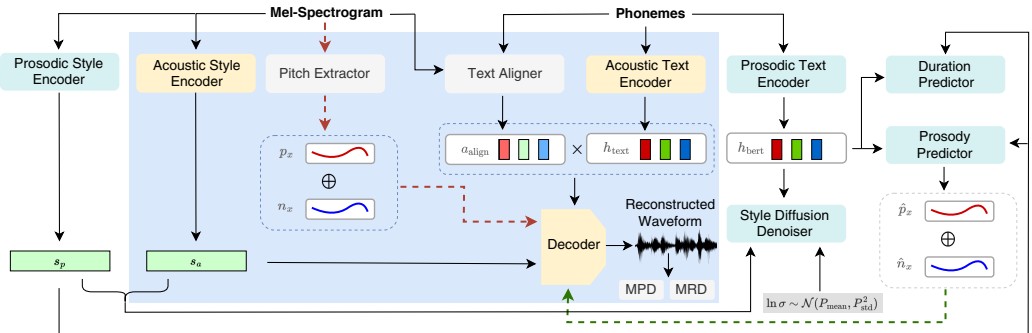

(a) Acoustic modules pre-training and joint training. To accelerate training, the pre-training first optimize modules inside the blue box; the joint training then follows to optimize all components except the pitch extractor, which is used to provide the ground truth label for pitch curves. The duration predictor is trained with only $\mathcal{L}_{\text{dur}}$.

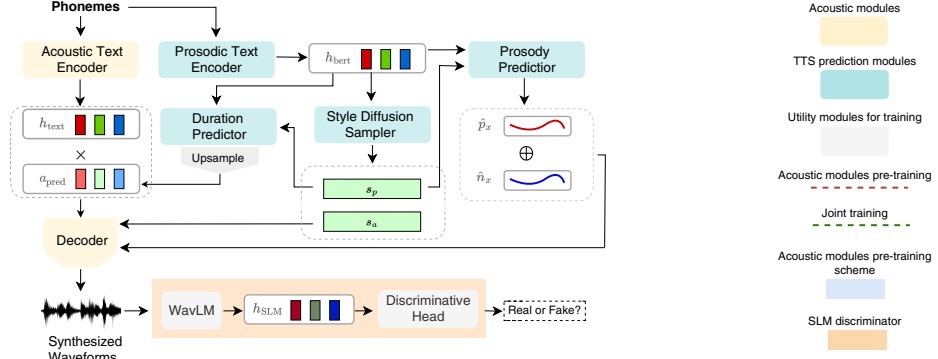

(b) SLM adversarial training and inference. WavLM is pre-trained and not tuned. Unlike (a), the duration predictor is trained E2E with all components using $\mathcal{L}_{slm}$ (eq. 5) via differentiable upsampling. This process is separate from (a) during training as the input texts can be different, but the gradients are accumulated for both processes in each batch to update the parameters.

Figure 1: Training and inference scheme of StyleTTS 2 for the single-speaker case. For the multi-speaker case, the acoustic and prosodic style encoders (denoted as $E$) first take reference audio $\boldsymbol{x}_{\text{ref}}$ of the target speaker and produce a reference style vector $\boldsymbol{c} = \boldsymbol{E}(\boldsymbol{x}_{\text{ref}})$. The style diffusion model then takes $\boldsymbol{c}$ as a reference to sample $\boldsymbol{s}_p$ and $\boldsymbol{s}_a$ that correspond to the speaker in $\boldsymbol{x}_{\text{ref}}$.

[47] along with the LSGAN loss functions [48] for decoder training, and incorporate the truncated pointwise relativistic loss function [49] to enhance sound quality (see Appendix F and G for details).

We found that well-trained acoustic modules, especially the style encoder, can accelerate the training process for TTS prediction modules. Therefore, before jointly optimizing all components, we first pre-train the acoustic modules along with the pitch extractor and text aligner via $\mathcal{L}_{\text{mel}}$, $\mathcal{L}_{\text{adv}}$, $\mathcal{L}_{\text{fm}}$ and TMA objectives for $N$ epochs where $N$ depends on the size of the training set, in the same way as the first training stage of [6]. However, we note that this pre-training is not an absolute necessity: despite being slower, starting joint training directly from scratch also leads to model convergence,

After acoustic module pre-training, we jointly optimize $\mathcal{L}_{\text{mel}}$, $\mathcal{L}_{\text{adv}}$, $\mathcal{L}_{\text{fm}}$, $\mathcal{L}_{\text{dur}}$, $\mathcal{L}_{\text{f0}}$ and $\mathcal{L}_{\text{n}}$, where $\mathcal{L}_{\text{mel}}$ is modified to match the mel-spectrograms of waveforms reconstructed from predicted pitch $\hat{p}_{\boldsymbol{x}}$ and energy $\hat{n}_{\boldsymbol{x}}$ (Fig 1a). During joint training, stability issues emerge from diverging gradients, as the style encoder must encode both acoustic and prosodic information. To address this inconsistency, we introduce a *prosodic* style encoder $E_p$ alongside the original *acoustic* style encoder $E_a$, previously denoted as $E$ in section 3.1. Instead of using $\boldsymbol{s}_a = E_a(\boldsymbol{x})$, predictors $S$ and $P$ take $\boldsymbol{s}_p = E_p(\boldsymbol{x})$ as the input style vector. The style diffusion model generates the augmented style vector $\boldsymbol{s} = [\boldsymbol{s}_p, \boldsymbol{s}_a]$. This modification effectively improves sample quality (see section 5.3). To further decouple the acoustic modules and predictors, we replace the phoneme representations $\boldsymbol{h}_{\text{text}}$ from $T$, now referred to as *acoustic* text encoder, with $\boldsymbol{h}_{\text{bert}}$ from another text encoder $B$ based on BERT transformers, denoted as *prosodic* text encoder. Specifically, we employ a phoneme-level BERT [7] pre-trained on extensive corpora of Wikipedia articles as the prosodic text encoder. This approach has been shown to enhance the naturalness of StyleTTS in the second stage [7], similar to our proposed usage here.

With differentiable upsampling and fast style diffusion, we can generate speech samples during training in a fully differentiable manner, just as during inference. These samples are used to optimize $\mathcal{L}_{slm}$ (eq. 5) during joint training to update the parameters of all components for inference (Fig 1b).

### 3.2.2 Style Diffusion

In StyleTTS 2, we model the speech $\boldsymbol{x}$ as a conditional distribution $p(\boldsymbol{x}|\boldsymbol{t}) = \int p(\boldsymbol{x}|\boldsymbol{t}, \boldsymbol{s}) p(\boldsymbol{s}|\boldsymbol{t}) \, d\boldsymbol{s}$ through a latent variable $\boldsymbol{s}$ that follows the distribution $p(\boldsymbol{s}|\boldsymbol{t})$. We call this variable the *generalized speech style*, representing any characteristic in speech beyond phonetic content $\boldsymbol{t}$, including but not limited to prosody, lexical stress, formant transitions, and speaking rate [6]. We sample $\boldsymbol{s}$ by EDM [50] that follows the combined probability flow [51] and time-varying Langevin dynamics [52]:

$$\boldsymbol{s} = \int -\sigma(\tau) \left[\beta(\tau)\sigma(\tau) + \dot{\sigma}(\tau)\right] \nabla_{\boldsymbol{s}} \log p_\tau(\boldsymbol{s}|\boldsymbol{t}) \, d\tau + \int \sqrt{2\beta(\tau)}\sigma(\tau) \, d\tilde{W}_\tau, \tag{1}$$

where $\sigma(\tau)$ is the noise level schedule and $\dot{\sigma}(\tau)$ is its time derivative, $\beta(\tau)$ is the stochasticity term, $\tilde{W}_\tau$ is the backward Wiener process for $\tau \in [T, 0]$ and $\nabla_{\boldsymbol{s}} \log p_\tau(\boldsymbol{s}|\boldsymbol{t})$ is the score function at time $\tau$.

We follow the EDM [50] formulation with the denoiser $K(\boldsymbol{s}; \boldsymbol{t}, \sigma)$ preconditioned as:

$$K(\boldsymbol{s}; \boldsymbol{t}, \sigma) := \left(\frac{\sigma_{\text{data}}}{\sigma^*}\right)^2 \boldsymbol{s} + \frac{\sigma \cdot \sigma_{\text{data}}}{\sigma^*} \cdot V\left(\frac{\boldsymbol{s}}{\sigma^*}; \boldsymbol{t}, \frac{1}{4}\ln\sigma\right), \tag{2}$$

where $\sigma$ follows a log-normal distribution $\ln\sigma \sim \mathcal{N}(P_{\text{mean}}, P_{\text{std}}^2)$ with $P_{\text{mean}} = -1.2$, $P_{\text{std}} = 1.2$, $\sigma^* := \sqrt{\sigma^2 + \sigma_{\text{data}}^2}$ is the scaling term, $\sigma_{\text{data}} = 0.2$ is the empirical estimate of the standard deviation of style vectors, and $V$ is a 3-layer transformer [53] conditioned on $\boldsymbol{t}$ and $\sigma$ which is trained with:

$$\mathcal{L}_{\text{edm}} = \mathbb{E}_{\boldsymbol{x}, \boldsymbol{t}, \sigma, \boldsymbol{\xi} \sim \mathcal{N}(0, I)} \left[\lambda(\sigma) \|K(\boldsymbol{E}(\boldsymbol{x}) + \sigma\boldsymbol{\xi}; \boldsymbol{t}, \sigma) - \boldsymbol{E}(\boldsymbol{x})\|_2^2\right], \tag{3}$$

where $\boldsymbol{E}(\boldsymbol{x}) := [E_a(\boldsymbol{x}), E_p(\boldsymbol{x})]$, and $\lambda(\sigma) := (\sigma^*/(\sigma \cdot \sigma_{\text{data}}))^2$ is the weighting factor. Under this framework, equation 16 becomes an ODE where the score function depends on $\sigma$ instead of $\tau$:

$$\frac{d\boldsymbol{s}}{d\sigma} = -\sigma\nabla_{\boldsymbol{s}} \log p_\sigma(\boldsymbol{s}|\boldsymbol{t}) = \frac{\boldsymbol{s} - K(\boldsymbol{s}; \boldsymbol{t}, \sigma)}{\sigma}, \qquad \boldsymbol{s}(\sigma(T)) \sim \mathcal{N}(0, \sigma(T)^2 I). \tag{4}$$

Unlike [50] that uses 2nd-order Heun, we solve eq. 4 with the ancestral DPM-2 solver [54] for fast and diverse sampling as we demand speed more than accuracy. On the other hand, we use the same scheduler as in [50] with $\sigma_{\min} = 0.0001, \sigma_{\max} = 3$ and $\rho = 9$. This combination allows us to sample a style vector for high-quality speech synthesis with only three steps, equivalent to running a 9-layer transformer model, minimally impacting the inference speed (see Appendix B for more discussions).

$V$ conditions on $\boldsymbol{t}$ through $h_{\text{bert}}$ concatenated with the noisy input $\boldsymbol{E}(\boldsymbol{x}) + \sigma\xi$, and $\sigma$ is conditioned via sinusoidal positional embeddings [53]. In the multispeaker setting, we model $p(\boldsymbol{s}|\boldsymbol{t}, \boldsymbol{c})$ by $K(\boldsymbol{s}; \boldsymbol{t}, \boldsymbol{c}, \sigma)$ with an additional speaker embedding $\boldsymbol{c} = \boldsymbol{E}(\boldsymbol{x}_{\text{ref}})$ where $\boldsymbol{x}_{\text{ref}}$ is the reference audio of the target speaker. The speaker embedding $\boldsymbol{c}$ is injected into $V$ by adaptive layer normalization [6].

### 3.2.3 SLM Discriminators

Speech language models (SLMs) encode valuable information ranging from acoustic to semantic aspects [55], and SLM representations are shown to mimic human perception for evaluating synthesized speech quality [45]. We uniquely transfer knowledge from SLM encoders to generative tasks via adversarial training by employing a 12-layer WavLM [11] pre-trained on 94k hours of data [1] as the discriminator. As the number of parameters of WavLM is greater than StyleTTS 2, to avoid discriminator overpowering, we fix the pre-trained WavLM model $W$ and append a convolutional neural network (CNN) $C$ as the discriminative head. We denote the SLM discriminator $D_{SLM} = C \circ W$. The input audios are downsampled to 16 kHz before being fed into $D_{SLM}$ to match that of WavLM. $C$ pools features $\boldsymbol{h}_{\text{SLM}} = W(\boldsymbol{x})$ from all layers with a linear map from $13 \times 768$ to $256$ channels. We train the generator components $(T, B, G, S, P, V,$ denoted as $\boldsymbol{G})$ and $D_{SLM}$ to optimize:

$$\mathcal{L}_{slm} = \min_{\boldsymbol{G}} \max_{D_{SLM}} \left(\mathbb{E}_{\boldsymbol{x}}[\log D_{SLM}(\boldsymbol{x})] + \mathbb{E}_{\boldsymbol{t}}[\log\left(1 - D_{SLM}(\boldsymbol{G}(\boldsymbol{t}))\right)]\right), \tag{5}$$

---

[1]Available at `https://huggingface.co/microsoft/wavlm-base-plus`

where $G(t)$ is the generated speech with text $t$, and $x$ is the human recording. [56] shows that:

$$D^*_{SLM}(x) = \frac{\mathbb{P}_{W \circ \mathcal{T}}(x)}{\mathbb{P}_{W \circ \mathcal{T}}(x) + \mathbb{P}_{W \circ \mathcal{G}}(x)}, \tag{6}$$

where $D^*_{SLM}(x)$ is the optimal discriminator, $\mathcal{T}$ and $\mathcal{G}$ represent true and generated data distributions, while $\mathbb{P}_{\mathcal{T}}$ and $\mathbb{P}_{\mathcal{G}}$ are their respective densities. The optimal $G^*$ is achieved if $\mathbb{P}_{W \circ \mathcal{T}} = \mathbb{P}_{W \circ \mathcal{G}}$, meaning that when converged, $G^*$ matches the generated and true distributions in the SLM representation space, effectively mimicking human perception to achieve human-like speech synthesis.

In equation 5, the generator loss is independent of ground truth $x$ and relies only on input text $t$. This enables training on out-of-distribution (OOD) texts, which we show in section 5.3 can improve the performance on OOD texts. In practice, to prevent $D_{SLM}$ from over-fitting on the content of the speech, we sample in-distribution and OOD texts with equal probability during training.

### 3.2.4 Differentiable Duration Modeling

The duration predictor produces phoneme durations $d_{\text{pred}}$, but the upsampling method described in section 3.1 to obtain $a_{\text{pred}}$ is not differentiable, blocking gradient flow for E2E training. NaturalSpeech [5] employs an attention-based upsampler [42] for human-level TTS. However, we find this approach unstable during adversarial training because we train our model using differentiable upsampling with only the adversarial objective described in eq. 5 and without extra loss terms due to the length mismatch caused by deviations of $d_{\text{pred}}$ from $d_{\text{gt}}$. Although this mismatch can be mitigated with soft dynamic time warping as used in [42, 5], we find this approach both computationally expensive and unstable with mel-reconstruction and adversarial objectives. To achieve human-level performance with adversarial training, a non-parametric upsampling method is preferred for stable training.

Gaussian upsampling [41] is non-parametric and converts the predicted duration $d_{\text{pred}}$ into $a_{\text{pred}}[n, i]$ using a Gaussian kernel $\mathcal{N}_{c_i}(n; \sigma)$ centered at $c_i := \ell_i - \frac{1}{2} d_{\text{pred}}[i]$ with the hyperparameter $\sigma$:

$$\mathcal{N}_{c_i}(n; \sigma) := \exp\left(-\frac{(n - c_i)^2}{2\sigma^2}\right), \quad (7) \qquad \ell_i := \sum_{k=1}^{i} d_{\text{pred}}[k], \tag{8}$$

where $\ell_i$ is the end position and $c_i$ is the center position of the $i^{\text{th}}$ phoneme $t_i$. However, Gaussian upsampling has limitations due to its fixed width of Gaussian kernels determined by $\sigma$. This constraint prevents it from accurately modeling alignments with varying lengths depending on $d_{\text{pred}}$. Non-attentive Tacotron [57] extends this by making $\sigma_i$ trainable, but the trained parameters introduce instability for E2E training with adversarial loss, similar to issues of attention-based upsamplers.

We propose a new non-parametric differentiable upsampler without additional training while taking into account the varying length of the alignment. For each phoneme $t_i$, we model the alignment as a random variable $a_i \in \mathbb{N}$, indicating the index of the speech frame the phoneme $t_i$ is aligned with. We define the duration of the $i^{\text{th}}$ phoneme as another random variable $d_i \in \{1, \ldots, L\}$, where $L = 50$ is the maximum phoneme duration hyperparameter, equivalent to 1.25 seconds in our setting. We observe that $a_i = \sum_{k=1}^{i} d_k$, but each $d_k$ is dependent on each other, making the sum difficult to model. Instead, we approximate $a_i = d_i + \ell_{i-1}$. The approximated probability mass function (PMF) of $a_i$ is

$$f_{a_i}[n] = f_{d_i + \ell_{i-1}}[n] = f_{d_i}[n] * f_{\ell_{i-1}}[n] = \sum_k f_{d_i}[k] \cdot \delta_{\ell_{i-1}}[n - k], \tag{9}$$

where $\delta_{\ell_{i-1}}$ is the PMF of $\ell_{i-1}$, a constant defined in eq. 8. This delta function is not differentiable, so we replace it with $\mathcal{N}_{\ell_{i-1}}$ defined in eq. 7 with $\sigma = 1.5$ (see Appendix C.2 for more discussions).

To model $f_{d_i}$, we modify the duration predictor to output $q[k, i]$, the probability of the $i^{\text{th}}$ phoneme having a duration of at least $k$ for $k \in \{1, \ldots, L\}$, optimized to be 1 when $d_{\text{gt}} \geq k$ with the cross-entropy loss (see Appendix G). Under this new scheme, we can approximate $d_{\text{pred}}[i] := \sum_{k=1}^{L} q[k, i]$, trained to match $d_{\text{gt}}$ with $\mathcal{L}_{\text{dur}}$ as in section 3.1. The vector $q[:, i]$ can be viewed as the unnormalized version of $f_{d_i}$, though it is trained to be uniformly distributed across the interval $[1, d_i]$. Since the number of speech frames $M$ is usually larger than the number of input phonemes $N$, this uniform distribution

aligns single phonemes to multiple speech frames as desired. Finally, we normalize the differentiable approximation $\tilde{f}_{a_i}[n]$ across the phoneme axis, as in [41], to obtain $\boldsymbol{a}_{\text{pred}}$ using the softmax function:

$$\boldsymbol{a}_{\text{pred}}[n, i] := \frac{e^{\left(\tilde{f}_{a_i}[n]\right)}}{\sum\limits_{i=1}^{N} e^{\left(\tilde{f}_{a_i}[n]\right)}}, \qquad (10) \qquad \tilde{f}_{a_i}[n] := \sum_{k=0}^{\hat{M}} q[n, i] \cdot \mathcal{N}_{\ell_{i-1}}(n - k; \sigma), \qquad (11)$$

where $\hat{M} := \lceil \ell_N \rceil$ is the predicted total duration of the speech and $n \in \{1, \ldots, \hat{M}\}$. An illustration of our proposed differentiable duration modeling is given in Figure 4 (Appendix C).

## 4 Experiments

### 4.1 Model Training

We performed experiments on three datasets: LJSpeech, VCTK, and LibriTTS. Our single-speaker model was trained on the LJSpeech dataset, consisting of 13,100 short audio clips totaling roughly 24 hours. This dataset was divided into training (12,500 samples), validation (100 samples), and testing (500 samples) sets, with the same split as [3, 5, 6]. The multispeaker model was trained on VCTK, comprising nearly 44,000 short clips from 109 native speakers with various accents. The data split was the same as [3], with 43,470 samples for training, 100 for validation, and 500 for testing. Lastly, we trained our model on the combined LibriTTS *train-clean-460* subset [14] for zero-shot adaptation. This dataset contains about 245 hours of audio from 1,151 speakers. Utterances longer than 30 seconds or shorter than one second were excluded. We distributed this dataset into training (98%), validation (1%), and testing (1%) sets, in line with [6]. The *test-clean* subset was used for zero-shot adaptation evaluation with 3-second reference clips. All datasets were resampled to 24 kHz to match LibriTTS, and the texts were converted into phonemes using phonemizer [58].

We used texts in the training split of LibriTTS as the out-of-distribution (OOD) texts for SLM adversarial training. We used iSTFTNet decoder for LJSpeech due to its speed and sufficient performance on this dataset, while the HifiGAN decoder was used for the VCTK and LibriTTS models. Acoustic modules were pre-trained for 100, 50, and 30 epochs on the LJSpeech, VCTK, and LibriTTS datasets, and joint training followed for 60, 40, and 25 epochs, respectively. We employed the AdamW optimizer [59] with $\beta_1 = 0, \beta_2 = 0.99$, weight decay $\lambda = 10^{-4}$, learning rate $\gamma = 10^{-4}$ and a batch size of 16 samples for both pre-training and joint training. The loss weights were adopted from [6] to balance all loss terms (see Appendix G for details). Waveforms were randomly segmented with a max length of 3 seconds. For SLM adversarial training, both the ground truth and generated samples were ensured to be 3 to 6 seconds in duration, the same as in fine-tuning of WavLM models for various downstream tasks [11]. Style diffusion steps were randomly sampled from 3 to 5 during training for speed and set to 5 during inference for quality. The training was conducted on four NVIDIA A40 GPUs.

### 4.2 Evaluations

We employed two metrics in our experiments: Mean Opinion Score of Naturalness (MOS-N) for human-likeness, and Mean Opinion Score of Similarity (MOS-S) for similarity to the reference for multi-speaker models. These evaluations were conducted by native English speakers from the U.S. on Amazon Mechanical Turk. All evaluators reported normal hearing and provided informed consent as monitored by the local institutional review board and in accordance with the ethical standards of the Declaration of Helsinki [2]. In each test, 80 random text samples from the test set were selected and converted into speech using our model and the baseline models, along with ground truth for comparison. Because [7] reported that many TTS models perform poorly for OOD texts, our LJSpeech experiments also included 40 utterances from Librivox spoken by the narrator of LJSpeech but from audiobooks not in the original dataset as the ground truth for OOD texts. To compare the difference between in-distribution and OOD performance, we asked the same raters to evaluate samples on both in-distribution and OOD texts.

Our baseline models consisted of the three highest-performing public models: VITS [3], StyleTTS [6], and JETS [32] for LJSpeech; and VITS, YourTTS [60], and StyleTTS for LibriTTS. Each synthesized

---

[2]We obtained approval for our protocol (number IRB-AAAR8655) from the Institutional Review Board.

Table 1: Comparative mean opinion scores of naturalness and similarity for StyleTTS 2 with p-values from Wilcoxon test relative to other models. Positive scores indicate StyleTTS 2 is better.

| Model | Dataset | CMOS-N (p-value) | CMOS-S (p-value) |
|---|---|---|---|
| Ground Truth | LJSpeech | $+$**0.28** $(p = 0.021)$ | — |
| NaturalSpeech | LJSpeech | $+$**1.07** $(p < 10^{-6})$ | — |
| Ground Truth | VCTK | $-0.02$ $(p = 0.628)$ | $+$**0.30** $(p = 0.081)$ |
| VITS | VCTK | $+$**0.45** $(p = 0.009)$ | $+$**0.43** $(p = 0.032)$ |
| Vall-E | LibriSpeech (zero-shot) | $+$**0.67** $(p < 10^{-3})$ | $-0.47$ $(p < 10^{-3})$ |

speech set was rated by 5 to 10 evaluators on a 1-5 scale, with increments of 0.5. We randomized the model order and kept their labels hidden, similar to the MUSHRA approach [61, 62]. We also conducted Comparative MOS (CMOS) tests to determine statistical significance, as raters can ignore subtle differences in MOS experiments [3, 7, 5]. Raters were asked to listen to two samples and rate whether the second was better or worse than the first on a -6 to 6 scale with increments of 1. We compared our model to the ground truth and NaturalSpeech [5] for LJSpeech, and the ground truth and VITS for VCTK. For the zero-shot experiment, we compared our LibriTTS model to Vall-E [8].

All baseline models, except for the publicly available VITS model on LibriTTS from ESPNet Toolkit [63], were official checkpoints released by the authors, including vocoders used in StyleTTS. As NaturalSpeech and Vall-E are not publicly available, we obtained samples from the authors and the official Vall-E demo page, respectively. For fairness, we resampled all audio to 22.5 kHz for LJSpeech and VCTK, and 16 kHz for LibriTTS, to match the baseline models. We conducted ablation studies using CMOS-N on LJSpeech on OOD texts from LibriTTS *test-clean* subset for more pronounced results as in [7]. For more details of our evaluation procedures, please see Appendix E.

## 5 Results

### 5.1 Model Performance

The results outlined in Table 1 establish that StyleTTS 2 outperforms NaturalSpeech by achieving a CMOS of $+1.07$ $(p \ll 0.01)$, setting a new standard for this dataset. Interestingly, StyleTTS 2 was favored over the ground truth with a CMOS of $+0.28$ $(p < 0.05)$. This preference may stem from dataset artifacts such as fragmented audiobook passages in the LJSpeech dataset that disrupt narrative continuity, thus rendering the ground truth narration seemingly unnatural. This hypothesis is corroborated by the performance of StyleTTS 2 on the VCTK dataset, which lacks narrative context, where it performs comparably to the ground truth (CMOS = $-0.02, p \gg 0.05$). Samples from our model were more similar to the reference audio speaker than the human recording, suggesting our model's effective use of the reference audio for style diffusion. Moreover, StyleTTS 2 scored higher than the previous state-of-the-art model VITS on VCTK, as evidenced by CMOS-N and CMOS-S.

Consistent with the CMOS results, our model achieved a MOS of 3.83, surpassing all previous models on LJSpeech (Table 2). In addition, all models, except ours, exhibited some degrees of quality degradation for out-of-distribution (OOD) texts. This corroborates the gap reported in [7], with our results providing additional ground truth references. On the other hand, our model did not show any degradation and significantly outperformed other models in MOS for OOD texts (Table 2), demonstrating its strong generalization ability and robustness towards OOD texts.

In zero-shot tests, StyleTTS 2 surpasses Vall-E in naturalness with a CMOS of $+0.67$ $(p \ll 0.01)$, although it falls slightly short in similarity (Table 1). Importantly, StyleTTS 2 achieved these results with only 245 hours of training data, compared to Vall-E's 60k hours, a 250x difference. This makes StyleTTS 2 a data-efficient alternative to large pre-training methods like Vall-E. The MOS results in Table 3 support these findings,

Table 2: Comparison of MOS with 95% confidence intervals (CI) on LJSpeech. $\text{MOS}_{\text{ID}}$ represents MOS-N for in-distribution texts, while $\text{MOS}_{\text{OOD}}$ is that for OOD texts.

| Model | $\text{MOS}_{\text{ID}}$ (CI) | $\text{MOS}_{\text{OOD}}$ (CI) |
|---|---|---|
| Ground Truth | 3.81 $(\pm 0.09)$ | 3.70 $(\pm 0.11)$ |
| StyleTTS 2 | **3.83** $(\pm$ **0.08**$)$ | **3.87** $(\pm$ **0.08**$)$ |
| JETS | 3.57 $(\pm 0.09)$ | 3.21 $(\pm 0.12)$ |
| VITS | 3.34 $(\pm 0.10)$ | 3.21 $(\pm 0.11)$ |
| StyleTTS + HifiGAN | 3.35 $(\pm 0.10)$ | 3.32 $(\pm 0.12)$ |

as our model exceeds all baseline models in both MOS-N and MOS-S. However, the difference in

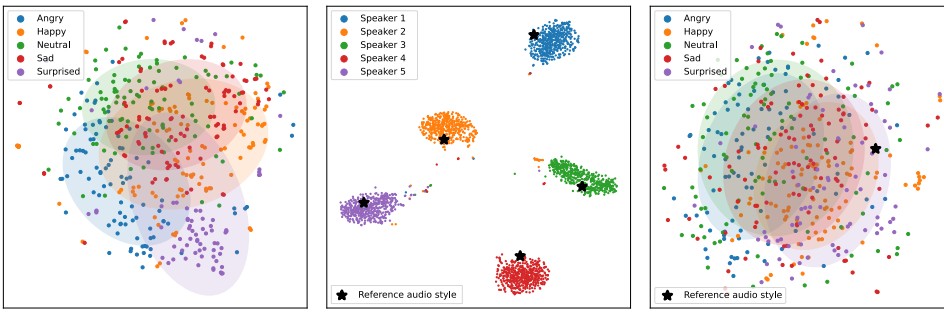

| (a) LJSpeech model. | (b) Unseen speakers on LibriTTS. | (c) Zoomed-in unseen speaker. |

Figure 2: t-SNE visualization of style vectors sampled via style diffusion from texts in five emotions, showing that emotions are properly separated for seen and unseen speakers. **(a)** Clusters of emotion from styles sampled by the LJSpeech model. **(b)** Distinct clusters of styles sampled from 5 unseen speakers by the LibriTTS model. **(c)** Loose clusters of emotions from Speaker 1 in (b).

MOS-S between StyleTTS and StyleTTS 2 was not statistically significant, hinting at future directions for improvement in speaker similarity.

## 5.2 Style Diffusion

Figure 2 shows t-SNE visualizations of style vectors created using our style diffusion process. Due to the lack of emotion-labeled audiobook text datasets, we used GPT-4 to generate 500 utterances across five emotions for this task [64]. In Figure 2a, style vectors from the LJSpeech model illustrate distinct emo-

Table 3: Comparison of MOS with 95% confidence intervals (CI) on *test-clean* subset of LibriTTS for zero-shot speaker adaptation.

| Model | MOS-N (CI) | MOS-S (CI) |
|---|---|---|
| Ground Truth | 4.60 ($\pm$ 0.09) | 4.35 ($\pm$ 0.10) |
| StyleTTS 2 | **4.15 ($\pm$ 0.11)** | **4.03 ($\pm$ 0.11)** |
| YourTTS | 2.35 ($\pm$ 0.07) | 2.42 ($\pm$ 0.09) |
| VITS | 3.69 ($\pm$ 0.12) | 3.54 ($\pm$ 0.13) |
| StyleTTS + HiFi-GAN | 3.91 ($\pm$ 0.11) | 4.01 ($\pm$ 0.10) |

tional styles in response to input text sentiment, demonstrating the model's capability to synthesize expressive speech in varied emotions without explicit emotion labels during training. This process was repeated with the LibriTTS model on five unseen speakers, each from a 3-second reference audio. As depicted in Figure 2b, distinct clusters form for each speaker, showcasing a wide stylistic diversity derived from a single reference audio. Figure 2c provides a more nuanced view of the first speaker, revealing visible emotion-based clusters despite some overlaps, indicating that we can manipulate the emotional tone of an unseen speaker regardless of the tones in the reference audio. These overlaps, however, can partly explain why the LibriTTS model does not perform as well as the LJSpeech model, as it is harder to disentangle texts from speakers in the zero-shot setting (see Appendix A.2 for more results). Table 4 displays our synthesized speech diversity against several baseline models with the coefficient of variation of duration ($CV_{dur}$) and pitch curve ($CV_{f0}$) from the same input text [3]. Our model yields the highest variation, indicating superior potential for generating diverse speech. Despite being diffusion-based, our model is faster than VITS, FastDiff [28], and ProDiff [18], two of the fastest diffusion-based TTS models, with 5 iterations of diffusion and iSTFTNet decoder.

## 5.3 Ablation Study

Table 4: Speech diversity metrics and real-time factor (RTF). ProDiff and FastDiff were tested with 4 steps of diffusion.

| Model | $CV_{dur}$ ↑ | $CV_{f0}$ ↑ | RTF ↓ |
|---|---|---|---|
| StyleTTS 2 | **0.0321** | **0.6962** | **0.0185** |
| VITS | 0.0214 | 0.5976 | 0.0599 |
| FastDiff | 0.0295 | 0.6490 | 0.0769 |
| ProDiff | 2e-16 | 0.5898 | 0.1454 |

Table 5: CMOS-N relative to the StyleTTS 2 baseline on OOD texts in the ablation study.

| Model | CMOS-N |
|---|---|
| w/o style diffusion | $-0.46$ |
| w/o differentiable upsampler | $-0.21$ |
| w/o SLM adversarial training | $-0.32$ |
| w/o prosodic style encoder | $-0.35$ |
| w/o OOD texts | $-0.15$ |

Table 5 details the ablation study, underlying the importance of our proposed components. When style vectors from style diffusion are substituted with randomly encoded ones as in [6], the CMOS is $-0.46$, highlighting the contribution of text-dependent style diffusion to achieving human-level TTS. Training without our differentiable upsampler and without the SLM discriminator results in a CMOS of $-0.21$ and $-0.32$, validating their key roles in natural speech synthesis. Removing the prosodic style encoder also yields a $-0.35$ CMOS. Last, excluding OOD texts from adversarial training leads to a CMOS of $-0.15$, proving its efficacy for improving OOD speech synthesis. Table 6 in Appendix A.3 shows similar outcomes with objective evaluations, further affirming the effectiveness of various components we proposed in this work. Figure 7 in Appendix D details a layer-wise analysis of input weights of the SLM discriminator, providing a different view of the efficacy of SLM discriminators.

## 6    Conclusions and Limitations

In this study, we present StyleTTS 2, a novel text-to-speech (TTS) model with human-level performance via style diffusion and speech language model discriminators. In particular, it exceeds the ground truth on LJSpeech and performs on par with it on the VCTK dataset. StyleTTS 2 also shows potential for zero-shot speaker adaption, with remarkable performance even on limited training data compared to large-scale models like Vall-E. With our innovative style diffusion method, StyleTTS 2 generates expressive and diverse speech of superior quality while ensuring fast inference time. While StyleTTS 2 excels in several areas, our results indicate room for improvement in handling large-scale datasets such as LibriTTS, which contain thousands of or more speakers, acoustic environments, accents, and other various aspects of speaking styles. The speaker similarity in the aforementioned zero-shot adaptation speaker task could also benefit from further improvements.

However, zero-shot speaker adaptation has the potential for misuse and deception by mimicking the voices of individuals as a potential source of misinformation or disinformation. This could lead to harmful, deceptive interactions such as theft, fraud, harassment, or impersonations of public figures that may influence political processes or trust in institutions. In order to manage the potential for harm, we will require users of our model to adhere to a code of conduct that will be clearly displayed as conditions for using the publicly available code and models. In particular, we will require users to inform those listening to samples synthesized by StyleTTS 2 that they are listening to synthesized speech or to obtain informed consent regarding the use of samples synthesized by StyleTTS 2 in experiments. Users will also be required to use reference speakers who have given consent to have their voice adapted, either directly or by license. Finally, we will make the source code publicly available for further research in speaker fraud and impersonation detection.

In addition, while human evaluators have favored StyleTTS 2 over ground truth with statistical significance on the LJSpeech dataset, this preference may be context-dependent. Original audio segments from larger contexts like audiobooks could inherently differ in naturalness when isolated, potentially skewing the evaluations in favor of synthesized speech. Additionally, the inherent variability in human speech, which is context-independent, might lead to lower ratings when compared to the more uniform output from StyleTTS 2. Future research should aim to improve evaluation methods to address these limitations and develop more natural and human-like speech synthesis models with longer context dependencies.

## 7    Acknowledgments

We thank Menoua Keshishian, Vishal Choudhari, and Xilin Jiang for helpful discussions and feedback on the paper. We also thank Grace Wilsey, Elden Griggs, Jacob Edwards, Rebecca Saez, Moises Rivera, Rawan Zayter, and D.M. for assessing the quality of synthesized samples and providing feedback on the quality of models during the development stage of this work. This work was funded by the national institute of health (NIHNIDCD) and a grant from Marie-Josee and Henry R. Kravis.

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

## Appendix A   Additional Evaluation Results

### A.1   Feedback Analysis from Survey Participants

On the last page of our survey, we encouraged the raters to provide their feedback on the study. While the majority opted not to comment, those who did provided valuable insights into our survey design and results interpretation. The main point of feedback revolved around the difficulty in discerning differences between different samples, specifically for the LJSpeech CMOS experiments. Below are a few representative comments from our CMOS experiment participants:

- Sounds great. At most points the differences were negligible.
- I really liked the person who was reading the clips.
  She sounds professional and was very good at changing her tone
  of voice. Sometimes it was hard to evaluate the clip because you
  had to listen carefully when they sounded similar.
  I didn't have any problems. I enjoyed participating Thank you!
- For most of the rounds, it was difficult to notice a difference
  between the voices.
- Interested to see the results - differences were sometimes
  noticeable, often subtle

This feedback strengthens our conclusion that StyleTTS 2 has reached human-like text-to-speech synthesis quality. Several participants offered specific observations regarding the differences:

- Some sounded a little unnatural because they sounded like an actor
  auditioning for a role - a little too dramatic.
- So, a couple of comments. I feel like the expressiveness of the
  speaker made me think it was more natural, but hyper expressiveness
  would be unnatural in certain contexts (and did feel odd with some of
  them but it felt more real). Additionally, the different voices, in
  being expressive, sometimes tended to turn into different emotions
  and contextual these emotions sometimes felt unnatural but felt real
  at the same time.
  Fun study though. I wish you all well!
- some sound relatively unexpected/unnatural, but most often it was
  just a minor difference

These comments suggest that the differences between StyleTTS 2 and the original audio mainly lie in subtle nuances like intonation and word emphasis.

Our results indicate that StyleTTS 2 scored statistically higher than the original audio in both MOS and CMOS, likely due to the absence of contextual continuity in the common evaluation procedure used in TTS studies. This disruption of narrative continuity in the LJSpeech dataset, which consists of isolated audiobook clips, might have led to the perceived unnaturalness of the ground truth compared to samples generated by StyleTTS 2. Future research should aim to incorporate context-aware long-form generation into human-like text-to-speech synthesis to improve evaluation fairness and relevance.

### A.2   Speech Expressiveness

Our model's ability to synthesize diverse and expressive speech is demonstrated by synthesizing speech from 500 text samples across five emotions generated using GPT-4. This reinforces the results visualized in Figure 2, which maps style vectors from texts associated with different emotions using t-SNE. We use the mean value of the F0 and energy curves to evaluate emotive speech.

As depicted in Figure 3 (a), speech synthesized by our model exhibits visible F0 and energy characteristics for each emotion, notably for anger and surprise texts, which deviate from the ground truth average. This substantiates our model's capacity to generate emotive speech from texts associated with specific emotions. Conversely, VITS shows certain degrees of insensitivity to emotional variance,

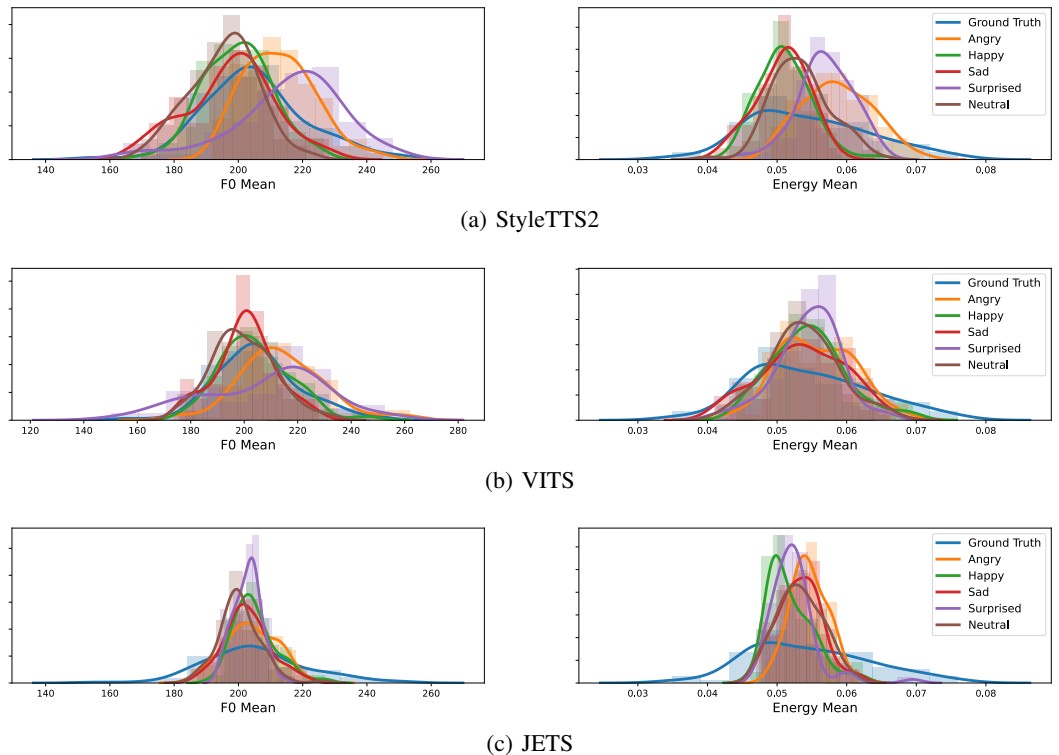

Figure 3: Histograms and kernel density estimation of the mean F0 and energy values of speech, synthesized with texts in five different emotions. The blue color ("Ground Truth") denotes the distributions of the ground truth samples in the test set. StyleTTS 2 shows distinct distributions for different emotions and produces samples that cover the entire range of the ground truth distributions.

as displayed in Figure 3 (b). JETS, shown in Figure 3 (c), fails to span the full distribution of F0 and energy, producing samples clustered around the mode. In contrast, our model extends across both F0 and energy distributions, approximating the range of the ground truth.

Since both StyleTTS 2 and VITS are probabilistic while JETS is deterministic, this shows that probabilistic models generate more expressive speech compared to deterministic ones. Moreover, our model demonstrates slightly better mode coverage compared to VITS at the right tail of the energy mean distribution, likely because we use diffusion-based models as opposed to the variational autoencoder used in VITS.

For audio samples of the synthesized speech and their corresponding emotional text, please refer to our demonstration page at `https://styletts2.github.io/#emo`.

## A.3 Ablation Study

The ablation study results, presented in Table 6, illustrate the importance of each component in our proposed model using several measures: mel-cepstral distortion (MCD), MCD weighted by Speech Length (SL) that evaluates both the length and the quality of alignment between two speeches (MCD-SL), root mean square error of log F0 pitch ($F_0$ RMSE), mean absolute deviation of phoneme duration (DUR MAD), and word error rate (WER), following [36]. The MCD and $F_0$ RMSE were computed between the samples generated by each model and the ground truth aligned with dynamic time warping. The MCD-SL was computed using a publicly available library [3]. The WER was computed with a pre-trained ASR model [4] from ESPNet [63].

Most notably, using randomly encoded style vectors instead of those sampled through style diffusion considerably affects all metrics, including the subjective CMOS score, thereby underscoring its

---

[3] `https://github.com/chenqi008/pymcd`
[4] Available at `https://zenodo.org/record/4541452`

Table 6: Comparison of models in ablation study with mel-cepstral distortion (MCD), MCD weighted by Speech Length (MCD-SL), root mean square error of log F0 pitch ($F_0$ RMSE), mean absolute deviation of phoneme duration (DUR MAD), and word error rate (WER) on LJSpeech. The CMOS results are copied directly from Table 5 as a reference.

| Model | MCD | MCD-SL | $F_0$ RMSE | DUR MAD | WER | CMOS |
|---|---|---|---|---|---|---|
| Proposed model | **4.93** | **5.34** | **0.651** | 0.521 | **6.50%** | **0** |
| w/o style diffusion | 8.30 | 9.33 | 0.899 | 0.634 | 8.77% | $-0.46$ |
| w/o SLM adversarial training | 4.95 | 5.40 | 0.692 | **0.513** | 6.52% | $-0.32$ |
| w/o prosodic style encoder | 5.04 | 5.42 | 0.663 | 0.543 | 6.92% | $-0.35$ |
| w/o differentiable upsampler | 4.94 | **5.34** | 0.880 | 0.525 | 6.54% | $-0.21$ |
| w/o OOD texts | **4.93** | 5.45 | 0.690 | 0.516 | 6.58% | $-0.15$ |

significance. Since [6] establishes that the style vectors crucially influence all aspects of the speech, such as pauses, emotions, speaking rates, and sound quality, style diffusion is the most important factor in producing speech close to the ground truth and natural human speech.

Excluding adversarial training with large speech language models (SLMs) results in a slight decline in MCD-SL and $F_0$ RMSE but does not impact WER. Interestingly, this version produces the lowest duration error, implying that SLM discriminators may cause minor underfitting on in-distribution texts, which can also be observed in models trained without OOD texts during SLM adversarial training. Nonetheless, subjective evaluations reveal a significant difference for out-of-distribution (OOD) texts for models trained without SLM discriminators (see Appendix D for more discussions).

Training without the differentiable upsampler increases the $F_0$ RMSE, but leaves the MCD, MCD-SL, DUR MAD, and WER unaffected. The removal of the prosodic style encoder affects all metric scores, thus underscoring its effectiveness in our model. Lastly, training without OOD texts only affects the $F_0$ RMSE in our objective evaluation.

Further details, discussions, and audio samples regarding these model variations can be found in Section 8 of our demo page: `https://styletts2.github.io/#ab`.

## Appendix B   Style Diffusion

### B.1   EDM Formulation

EDM [50] formulates the sampling procedure of diffusion-based generative model as an ordinary differential equation (ODE) (Eq. 1 in [50]):

$$dx = -\dot{\sigma}(t)\sigma(t)\nabla_x \log p(x; \sigma(t))\, dt, \tag{12}$$

where the noise level schedule and its derivative are represented by $\sigma(t)$ and $\dot{\sigma}(t)$, respectively. The score function of $p(x)$ is $\nabla_x \log p(x; \sigma(t))$, and $p(x)$ is the targeted data distribution (please refer to Appendix B in [50] for a detailed derivation). The equation can be rewritten in terms of $d\sigma$ as:

$$dx = -\sigma \nabla_x \log p(x; \sigma)\, d\sigma. \tag{13}$$

The denoising score matching objective is defined as follows (Eq. 108 in [50]):

$$\mathbb{E}_{y \sim p_{\text{data}}, \xi \sim \mathcal{N}(0, \sigma^2 I), \ln \sigma \sim \mathcal{N}(P_{\text{mean}}, P_{\text{std}}^2)} \left[ \lambda(\sigma) \left\| D(y + \xi; \sigma) - y \right\|_2^2 \right]. \tag{14}$$

Here, $y$ denotes the training sample from data distribution $p_{\text{data}}$, $\xi$ is the noise, and $D$ is the denoiser, defined as (Eq. 7 in [50]):

$$D(x; \sigma) := c_{\text{skip}}(\sigma)x + c_{\text{out}}(\sigma)F_\theta(c_{\text{in}}(\sigma)x; c_{\text{noise}}(\sigma)). \tag{15}$$

In the above equation, $F_\theta$ represents the trainable neural network, $c_{\text{skip}}(\sigma)$ modulates the skip connection, $c_{\text{in}}(\sigma)$ and $c_{\text{out}}(\sigma)$ regulate input and output magnitudes respectively, and $c_{\text{noise}}(\sigma)$ maps

noise level $\sigma$ into a conditioning input for $F_\theta$. With this formulation, the score function is given by (Eq. 3 in [50]):

$$\nabla_{\boldsymbol{x}} \log p(\boldsymbol{x}; \sigma(t)) = (D(\boldsymbol{x}; \sigma) - \boldsymbol{x}) / \sigma^2. \tag{16}$$

In Table 1 of [50], the authors define the scaling factors as follows:

$$c_{\text{skip}}(\sigma) := \sigma_{\text{data}}^2 / (\sigma^2 + \sigma_{\text{data}}^2) = (\sigma_{\text{data}}/\sigma^*)^2, \tag{17}$$

$$c_{\text{out}}(\sigma) := \sigma \cdot \sigma_{\text{data}} / \sqrt{\sigma_{\text{data}}^2 + \sigma^2} = \sigma \cdot \sigma_{\text{data}}/\sigma^*, \tag{18}$$

$$c_{\text{in}}(\sigma) := 1 / \sqrt{\sigma_{\text{data}}^2 + \sigma^2} = 1/\sigma^*, \tag{19}$$

$$c_{\text{noise}}(\sigma) := \frac{1}{4} \ln(\sigma), \tag{20}$$

where $\sigma^* := \sqrt{\sigma_{\text{data}}^2 + \sigma^2}$, a shorthand notation for simplicity.

Inserting equations 17, 18, 19 and 20 into equation 15 yields:

$$
\begin{aligned}
D(\boldsymbol{x}; \sigma) &= c_{\text{skip}}(\sigma)\boldsymbol{x} + c_{\text{out}}(\sigma) F_\theta(c_{\text{in}}(\sigma)\boldsymbol{x}; c_{\text{noise}}(\sigma)) \\
&= \left(\frac{\sigma_{\text{data}}}{\sigma^*}\right)^2 \boldsymbol{x} + \frac{\sigma \cdot \sigma_{\text{data}}}{\sigma^*} \cdot F_\theta\left(\frac{\boldsymbol{x}}{\sigma^*}; \frac{1}{4}\ln \sigma\right),
\end{aligned}
\tag{21}
$$

Combining equation 13 and 16, we obtain

$$
\begin{aligned}
\frac{d\boldsymbol{x}}{d\sigma} &= -\sigma \nabla_{\boldsymbol{x}} \log p(\boldsymbol{x}; \sigma(t)) \\
&= -\sigma \left(D(\boldsymbol{x}; \sigma) - \boldsymbol{x}\right)/\sigma^2 \\
&= \frac{\boldsymbol{x} - D(\boldsymbol{x}; \sigma)}{\sigma}.
\end{aligned}
\tag{22}
$$

Equations 2 and 4 can be recovered by replacing $\boldsymbol{x}$ with $\boldsymbol{s}$, $p(\boldsymbol{x})$ with $p(\boldsymbol{s}|\boldsymbol{t})$, $F_\theta$ with $V$ and $D$ with $K$ in equations 21 and 22 respectively.

In EDM, the time step $\{\sigma_0, \sigma_1, \ldots, \sigma_{N-1}\}$ for the total step $N$ is defined as (Eq. 5 in [50]):

$$\sigma_{i<N} := \left(\sigma_{\text{max}}^{\frac{1}{\rho}} + \frac{i}{N-1}\left(\sigma_{\text{min}}^{\frac{1}{\rho}} - \sigma_{\text{max}}^{\frac{1}{\rho}}\right)\right)^{\rho}, \tag{23}$$

where $\sigma_{\text{max}} = \sigma_0$, $\sigma_{\text{min}} = \sigma_{N-1}$, and $\rho$ is the factor that shortens the step lengths near $\sigma_{\text{min}}$ at the expense of longer steps near $\sigma_{\text{max}}$. Optimal performance is observed in [50] when $\sigma_{\text{max}} \gg \sigma_{\text{data}}$ and $\rho \in [5, 10]$. Through empirical tests, we set $\sigma_{\text{min}} = 0.0001, \sigma_{\text{max}} = 3$ and $\rho = 9$ in our work, allowing fast sampling with small step sizes while producing high-quality speech samples.

## B.2 Effects of Diffusion Steps

Table 7: Comparision of mel-cepstral distortion (MCD), MCD weighted by Speech Length (MCD-SL), root mean square error of log F0 pitch ($F_0$ RMSE), word error rate (WER), real-time factor (RTF), coefficient of variation of duration ($CV_{\text{dur}}$), and coefficient of variation of pitch ($CV_{\text{f0}}$) between different diffusion steps.

| Step | MCD $\downarrow$ | MCD-SL $\downarrow$ | $F_0$ RMSE $\downarrow$ | WER $\downarrow$ | RTF $\downarrow$ | $CV_{\text{dur}}$ $\uparrow$ | $CV_{\text{f0}}$ $\uparrow$ |
|---|---|---|---|---|---|---|---|
| 4 | **4.90** | 5.34 | **0.650** | 6.72% | **0.0179** | 0.0207 | 0.5473 |
| 8 | 4.93 | 5.33 | 0.674 | 6.53% | 0.0202 | 0.0466 | 0.7073 |
| 16 | 4.92 | 5.34 | 0.665 | **6.44%** | 0.0252 | **0.0505** | 0.7244 |
| 32 | 4.92 | **5.32** | 0.663 | 6.56% | 0.0355 | 0.0463 | **0.7345** |
| 64 | 4.91 | 5.34 | 0.654 | 6.67% | 0.0557 | 0.0447 | 0.7245 |
| 128 | 4.92 | 5.33 | 0.656 | 6.73% | 0.0963 | 0.0447 | 0.7256 |

The impact of diffusion steps on sample quality, speed, and diversity was studied using multiple indicators. We assessed the sample quality using mel-cepstral distortion (MCD), MCD weighted by

Speech Length (MCD-SL), root mean square error of log F0 pitch ($F_0$ RMSE), and word error rate (WER). We also examined the influence of diffusion steps on computational speed using the real-time factor (RTF), which was computed on a single Nvidia RTX 2080 Ti GPU. Given the application of the ancestral solver, which introduces noise at each integration step [52], it was postulated that a higher number of steps would lead to more diverse samples. This hypothesis was tested using the coefficient of variation on the F0 curve ($CV_{f0}$) and duration ($CV_{dur}$).

The model was run with diffusion steps ranging from 4 to 128, as results with only two steps were disregarded due to the characteristics of the noise scheduler in EDM. Here, $\sigma_0 = \sigma_{max}$ and $\sigma_{N-1} = \sigma_{min}$, and therefore, with merely two diffusion steps, the attainment of samples of acceptable quality is not feasible.

The results presented in Table 7 reveal negligible disparities in sample quality across varying diffusion steps, with satisfactory quality samples being producible with as few as three steps in our experiments. Consequently, we decided to randomly sample styles with diffusion steps ranging from 3 to 5 during training in order to save time and GPU RAM. Furthermore, we observed an incremental rise in speech diversity relative to the number of diffusion steps, reaching a plateau around 16 steps. Thereafter, the increase is marginal, with a slight decrease in diversity noticed when the steps are large, potentially attributable to the refining of the noise schedule with higher step counts. This could cause the ancestral solver to converge to a fixed set of solutions despite the noise introduced at each step.

Notably, optimal results in terms of sample quality and diversity, paired with acceptable computational speed, were achieved with around 16 diffusion steps. Even though the RTF experienced a 30% increase compared to 4 steps, it still outperformed VITS by twice the speed, making it a suitable choice for real-time applications.

## B.3 Consistent Long-Form Generation

---
**Algorithm 1** Long-form generation with style diffusion

---
    **procedure** LONGFORM($t, \alpha$)          ▷ $t$ is the input paragraph, $\alpha \in [0, 1]$ is the weighting factor
        $T \leftarrow$ SPLIT($t$)          ▷ Split $t$ into $N$ sentences $\{T_0, T_1, \ldots, T_{N-1}\}$
        **for** $i \in \{0, \ldots, N-1\}$ **do**
            $s_{curr} \leftarrow$ STYLEDIFFUSION($T_i$)         ▷ Sample a style vector with the current text $T_i$
            **if** $s_{prev}$ **then**         ▷ Check if the previous style vector is defined
                $s_{curr} \leftarrow \alpha s_{curr} + (1 - \alpha)s_{prev}$         ▷ Convex combination interpolation
            **end if**
            $x_i \leftarrow$ SYNTHESIZE($T_i, s_{curr}$)         ▷ Synthesize with the interpolated $s_{curr}$
            $s_{prev} \leftarrow s_{curr}$         ▷ Set $s_{prev}$ for the next iteration
        **end for**
        **return** CONCAT($\{x_0, x_1, \ldots, x_{N-1}\}$)         ▷ Return concatenated speech from all sentences
    **end procedure**

---

Our findings indicate that style diffusion creates significant variation in samples, a characteristic that poses challenges for long-form synthesis. In this scenario, a long paragraph is usually divided into smaller sentences for generation, sentence by sentence, in the same way as real-time applications. Using an independent style for each sentence may generate speech that appears inconsistent due to differences in speaking styles. Conversely, maintaining the same style from the first sentence throughout the entire paragraph results in monotonic, unnatural, and robotic-sounding speech.

We empirically observe that the latent space underlying the style vectors generally forms a convex space. Consequently, a convex combination of two style vectors yields another style vector, with the speaking style somewhere between the original two. This allows us to condition the style of the current sentence on the previous sentence through a simple convex combination. The pseudocode of this algorithm, which uses interpolated style vectors, is provided in Algorithm 1.

An example of a long paragraph generated using this method with $\alpha = 0.7$ is available on our demo page: `https://styletts2.github.io/#long`.

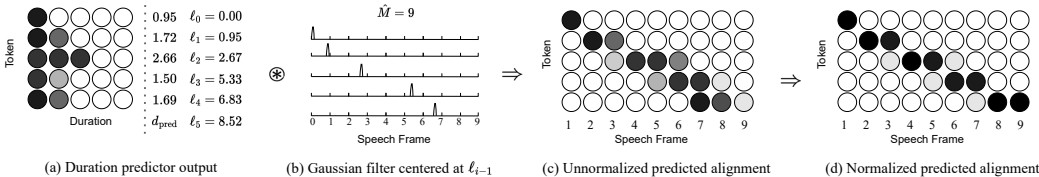

|  | (a) Duration predictor output | (b) Gaussian filter centered at $\ell_{i-1}$ | (c) Unnormalized predicted alignment | (d) Normalized predicted alignment |

Figure 4: Illustration of our proposed differentiable duration upsampler. **(a)** Probability output from the duration predictor for 5 input tokens with $L = 5$ . **(b)** Gaussian filter $\mathcal{N}_{\ell_{i-1}}$ centered at $\ell_{i-1}$. **(c)** Unnormalized predicted alignment $\tilde{f}_{a_i}[n]$ from the convolution operation between (a) and (b). **(d)** Normalized predicted alignment $\boldsymbol{a}_{\text{pred}}$ over the phoneme axis.

## B.4   Style Transfer

Our model, contrary to traditional end-to-end (E2E) TTS systems that directly map texts to speech-related representations, decouples speech content and style via a style vector. This setup enables the transfer of a style sourced from a text with specific emotions or stylistic nuances to any given input text. The procedure involves generating a style vector from a text reflecting desired aspects of the target style (e.g., emotions, speaking rates, recording environments, etc.) and synthesizing speech from any input text using this style vector. This process also extends to unseen speakers, as demonstrated on our demo page at `https://styletts2.github.io/#libri`. It is important to note that the relationship between speech style and input text is learned in a self-supervised manner without manual labeling, analogous to the CLIP embedding used in Stable Diffusion.

# Appendix C   Differentiable Duration Modeling

## C.1   DSP-Based Formulation

In section 3.2.4, we formulated the differentiable duration model using probability theory. This section provides an alternative formulation based on principles of digital signal processing (DSP). The upsampler is formulated using the following DSP properties:

$$x[n] * \delta_k[n] = x[n+k], \qquad (24) \qquad\qquad \delta_c(x) = \lim_{\sigma \to 0} \frac{1}{\sigma\sqrt{2\pi}} \mathcal{N}_c(x; \sigma). \qquad (25)$$

Here, $\delta_k[n]$ is the Kronecker delta function, $\delta_c(x)$ is the Dirac delta function, and $\mathcal{N}_c(x; \sigma)$ is the Gaussian kernel centered at $c$, as defined in eq. 7. These properties state that a delta function, $\delta_k$, through convolution operation, shifts the value of a function to a new position by $k$, and a properly normalized Gaussian kernel converges to the delta function as $\sigma \to 0$. Therefore, by properly selecting $\sigma$, we can approximate the shifting property of the delta function with $\mathcal{N}_c(x; \sigma)$ and adjust the duration predictor output $q[:, i]$ to the starting position $\ell_{i-1}$ of the $i^{\text{th}}$ phoneme. Figure 4 illustrates our proposed differentiable duration modeling, and a real example of the duration predictor output, the predicted alignment using the non-differentiable upsampling method described in section 3.1, and the alignment using our proposed method is given in Figure 5.

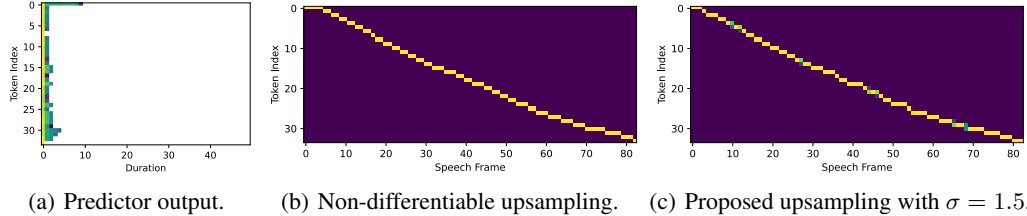

|  | (a) Predictor output. | (b) Non-differentiable upsampling. | (c) Proposed upsampling with $\sigma = 1.5$. |

Figure 5: An example of duration predictor output and the predicted alignment with and without differentiable duration upsampler. (a) displays the log probability from the duration predictor for improved visualization. Although (b) and (c) differs, the duration predictor is trained end-to-end with the SLM discriminator, making the difference perceptually indistinguishable in synthesized speech.

## C.2 Effects of $\sigma$ on Training Stability and Sample Quality

The gradient propagation from the SLM discriminator to the duration predictor is crucial for our model's performance. The norm of this gradient, computed using the chain rule, is the product of the norms of individual functions' gradients. The magnitude of the partial derivatives of $\mathcal{N}_c(x;\sigma)$ is:

$$\left|\frac{\partial \mathcal{N}_c(x;\sigma)}{\partial x}\right| = \left|\frac{\partial \mathcal{N}_c(x;\sigma)}{\partial c}\right| = \left|\pm\frac{(x-c)}{\sigma^2}\mathcal{N}_c(x;\sigma)\right| = \frac{|x-c|}{\sigma^2}|\mathcal{N}_c(x;\sigma)| \leq \frac{C}{\sigma^2}, \quad (26)$$

where $C$ is a constant dependent on $L$. The gradient norm is thus of order $O(1/\sigma^k)$, where $k$ depends on the input dimension. As $\sigma \to 0$, $\mathcal{N}_c(x;\sigma)$ approaches $\delta_c(x)$, but the gradient norm $\|\nabla \mathcal{N}_c(x;\sigma)\|$ goes to infinity. Moreover, smaller $\sigma$ values may not yield better $\delta$ approximations due to numerical instability, particularly when dealing with small values such as the predictor output $q \in [0,1]$, which is multiplied by $\mathcal{N}_c(x;\sigma)$ in eq. 11, exacerbating numerical instability.

We investigate the trade-off by examining the impact of $\sigma$ on training stability and sample quality. Training stability is represented by the gradient norm from the SLM discriminator to the duration predictor, and sample quality is measured by the mel cepstral distortion (MCD) between speech synthesized using the predicted alignment generated through the non-differentiable upsampling method (section 3.1) and our proposed differentiable upsampling method. We use the maximum gradient over an epoch to represent training stability, as large gradient batches often cause gradient explosion and unstable training. We compute these statistics for $\sigma$ values from 0.01 to 10.0 on a log scale, with results shown in Figure 6.

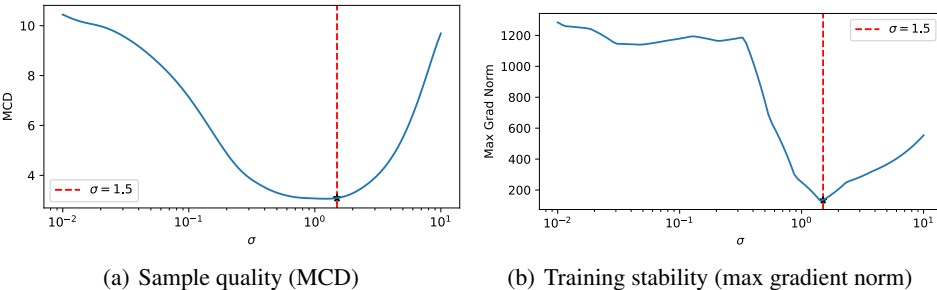

(a) Sample quality (MCD)  (b) Training stability (max gradient norm)

Figure 6: Effects of $\sigma$ on MCD and max gradient norm. Our choice of $\sigma = 1.5$ is marked with a star symbol. **(a)** MCD between samples synthesized with differentiable and non-differentiable upsampling over different $\sigma$. **(b)** The maximum norm of gradients from the SLM discriminator to the duration predictor over an epoch of training with different $\sigma$.

Our chosen $\sigma = 1.5$ minimizes both MCD and gradient norm across a wide range of $\sigma$ values, making it an optimal choice for upsampling. This value also aligns with the typical phoneme duration of 2 to 3 speech frames, as Gaussian kernels with $\sigma = 1.5$ span approximately 3 speech frames.

# Appendix D   SLM Discriminators

## D.1   Layer-Wise Analysis

Following the approach of the WavLM study [11], we performed a layer-wise feature importance analysis for the speech language model (SLM) discriminator. The weights of the input linear projection layer into the convolutional discriminative head were normalized using the average norm values of WavLM features for samples in the test set across LJSpeech, VCTK, and LibriTTS datasets. Figure 7 depicts the normalized weight magnitudes across layers.

In the LJSpeech and LibriTTS models, the initial (1 and 2) and middle layers (6 and 7) showed the highest importance, while the final layers (10 and 11) were of diminishing importance. The least influential was the final layer, consistent with previous findings that ascribe the primary role of this layer to pretext tasks over downstream tasks [55, 10, 11]. Notably, layers 0, 1, and 2, which encode acoustic information like energy, pitch, and signal-to-noise ratio (SNR), emerged as the most

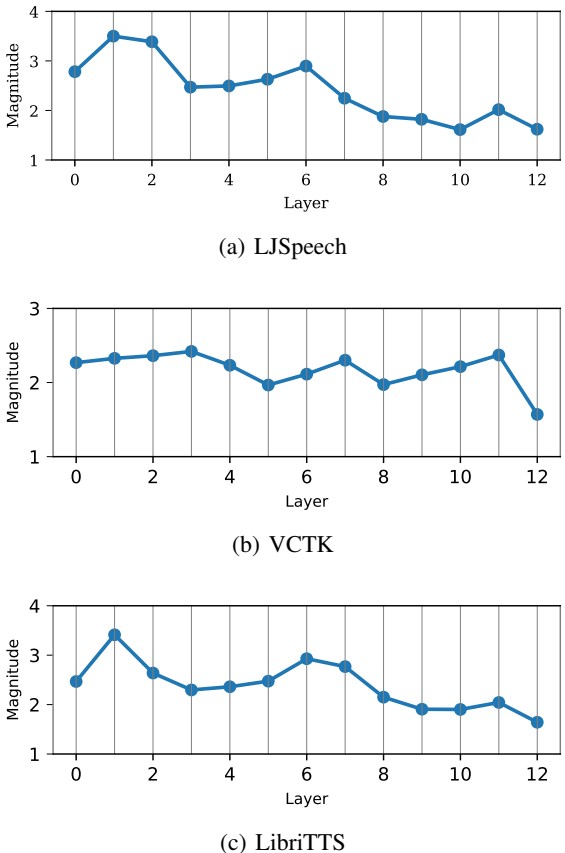

(a) LJSpeech

(b) VCTK

(c) LibriTTS

Figure 7: Layer-wise input weight magnitude to the SLM discriminators across different datasets. The layer importance shows a divergent pattern for the VCTK model relative to the LJSpeech and LibriTTS models, showcasing the impact of contextual absence on the SLM discriminators.

crucial. Layers 5, 6, and 7, encoding semantic aspects like word identity and meaning [55], held secondary importance. This indicates that the SLM discriminator learns to fuse acoustic and semantic information to derive paralinguistic attributes, such as prosody, pauses, intonations, and emotions, which are critical in distinguishing real and synthesized speech. The outcome reaffirms the SLM discriminator's capacity to enhance the emotional expressiveness and prosody of synthesized speech.

In contrast, the SLM discriminator for the VCTK dataset demonstrated no distinct layer preference, likely due to its limited contextual information. Unlike the storytelling and audiobook narrations in the LJSpeech and LibriTTS datasets, the VCTK dataset only involves reading a standard paragraph devoid of specific context or emotions. This context shortage could explain the marginal performance increase of our model over VITS on the VCTK dataset compared to LJSpeech and LibriTTS datasets (see Tables 1, 2, 3), as the advantages of style diffusion and SLM adversarial training in our model are less pronounced in datasets with restricted expressiveness and emotional variety.

### D.2 Training Stability

Figure 6(b) shows that the maximum norm gradient from the SLM discriminator to the duration predictor can reach up to 200 even with $\sigma = 1.5$, potentially destabilizing training, given that the gradient norms for other modules are generally less than 10. This instability is particularly concerning when the gradient is back-propagated to the prosodic text encoder, which employs a BERT model that is notably sensitive to gradient explosion. To mitigate this issue, we adopt a strategy for scaling the gradient from the SLM discriminator to the duration predictor, a common practice to prevent gradient explosion in GAN training [31].

We found that the gradient norm of the entire predictor typically falls between 10 and 20. As such, we implement a scaling factor of 0.2 to the gradient norm when it surpasses 20. Additionally, we also account for potential instabilities arising from the sigmoid function used by the duration predictor for probability output. We mitigate these by scaling the gradient to the last projection layer and the LSTM layer within the duration predictor by a factor of 0.01. These scaling values have consistently shown stability across different datasets and can be modified to accommodate dataset-specific attributes. All the models presented in this paper were trained with these scaling factors.

## Appendix E   Subjective Evaluation Procedures

In this section, we propose a detailed and standardized evaluation procedure and elucidate our adherence to these guidelines in the current work.

### E.1   Proposed Framework for Subjective Evaluation

In light of the findings in [43] along with guidelines from [5], we recommend the following procedures for assessing human-like text-to-speech models:

1. Raters should be native speakers of the language used in the model's training corpus, with provisions in place to exclude falsely claimed native speakers.

2. Attention checks should be incorporated into the evaluation to ensure thoughtful, careful completion, and raters who do not pass the attention checks should be excluded from the outcome analysis.

3. The objectives of the evaluation should be clearly stated. For instance, when assessing naturalness, its definition should be explicit within the evaluation framework.

4. Ideally, all models used in the experiments should be publicly available, and official implementations are preferred over unofficial implementations. If no official implementations are publicly available, the testing samples can be requested from the authors of the papers, or from implementations that have been employed in other papers.

5. For model comparisons, the MUSHRA-based approach, where paired samples from all models are presented, should be employed, in contrast to the traditional MOS evaluation scheme that presents model samples in isolation.

6. For nuanced model comparison and determining statistical significance, CMOS should be used in place of the MOS and MUSHRA evaluations. Each sample should be rated by at least 20 raters with more than 50 samples to claim statistical significance compared to ground truth to assess human-level quality [5].

### E.2   Evaluation Details

The first criterion was ensured by selecting native English-speaking participants residing in the United States. Upon launching our survey on MTurk, we activated the following filters:

- HIT Approval Rate (%) for all Requesters' HITS: `greater than 95`.
- Location: `is UNITED STATES (US)`.
- Number of HITs Approved: `greater than 50`.

For each batch, completed responses were collected exclusively from self-identified native speakers in the United States, verified by residential IP addresses (i.e., not proxy or VPN) with an online service [5].

To fulfill the second criterion, attention checks were implemented differently for MOS and CMOS evaluations. In the MOS assessment, we utilized the average score given by a participant to ground truth audios, unbeknownst to the participants, to ascertain their attentiveness. We excluded ratings from those whose average score for the ground truth did not rank in the top three among all five models. In the CMOS evaluation, we assessed the consistency of the rater's scoring; if the score's

---

[5] Avaiable at `https://www.ipqualityscore.com/free-ip-lookup-proxy-vpn-test`

sign (indicating whether A was better or worse than B) differed for over half the sample set (in our case, 10 samples), the rater was disqualified. Six raters were eliminated through this process in all of our experiments.

The third criterion was ensured by providing explicit definitions of "naturalness" and "similarity" within our surveys, exemplified by the following lines (see our survey links [6][7][8] for more detail):

- Naturalness:

  ```
  Some of them may be synthesized while others may be spoken by an
  American audiobook narrator.

  Rate how natural each audio clip sounds on a scale of 1 to 5 with
  1 indicating completely unnatural speech (bad) and 5 completely
  natural speech (excellent).

  Here, naturalness includes whether you feel the speech is spoken
  by a native American English speaker from a human source.
  ```
- Similarity:

  ```
  Rate whether the two audio clips could have been produced by the
  same speaker or not on a scale of 1 to 5 with 1 indicating
  completely different speakers and 5 indicating exactly the same
  speaker.

  Some samples may sound somewhat degraded/distorted; for this
  question, please try to listen beyond the distortion of the
  speech and concentrate on identifying the voice (including the
  person's accent and speaking habits, if possible).
  ```

In accordance with the fourth criterion, official model checkpoints were utilized in both MOS and CMOS evaluations. Specifically, for the MOS evaluation of LJSpeech, checkpoints for VITS[9], JETS[10], and StyleTTS with HifiGAN vocoder[11] were sourced directly from the authors. Likewise, for the MOS evaluation on LibriTTS, official checkpoints for StyleTTS with the HifiGAN vocoder and YourTTS (utilizing the Exp 4 checkpoint)[12] were used. Since there is no official checkpoint of VITS for LibriTTS, we used the VITS model from ESPNet [13] on LibriTTS, which was previously employed as a baseline model in [6]. For the CMOS experiment on VCTK with VITS, we used official checkpoints and implementation. For the CMOS evaluation with NaturalSpeech, we obtained 20 demo samples directly from the author. Regarding Vall-E's CMOS experiment, we collected 30 samples from LibriTTS, along with their corresponding 3-second prompts from the official demo page[14].

Fulfilling the fifth criterion, we employed a MUSHRA-based approach in our MOS evaluations. Lastly, we assured that 20 raters evaluated each sample in our CMOS experiments, after excluding ineligible raters. For NaturalSpeech and Vall-E, where additional samples were not available, we compensated by doubling the number of raters.

## Appendix F  Detailed Model Architectures

This section offers a comprehensive outline of the enhanced StyleTTS 2 architecture. The model integrates eight original StyleTTS modules with the addition of an extra style diffusion denoiser, prosodic style encoder, and prosodic text encoder.

---

[6]LJSpeech MOS: `https://survey.alchemer.com/s3/7376864/MOS-MIX-ID-OOD-B1`

[7]LJSpeech CMOS: `https://survey.alchemer.com/s3/7332116/CMOS-gt-styletts2-0430-b4`

[8]LibriTTS MOS: `https://survey.alchemer.com/s3/7387883/MOS-LibriTTS-0508-b2`

[9]`https://github.com/jaywalnut310/vits`

[10]`https://huggingface.co/imdanboy/jets`

[11]`https://github.com/yl4579/StyleTTS`

[12]`https://github.com/Edresson/YourTTS`

[13]Available at `https://huggingface.co/espnet/kan-bayashi_libritts_xvector_vits`

[14]`https://www.microsoft.com/en-us/research/project/vall-e-x/vall-e/`

The model keeps the same architecture for the acoustic text encoder, text aligner, and pitch extractor as in the original StyleTTS [6]. The architecture of both the acoustic style encoder and prosodic style encoder follows that of the style encoder from StyleTTS [6]. The prosodic text encoder is a pre-trained PL-BERT[15] model [7]. Additionally, we adopt the same discriminators, MPD and MRD, as specified in [31]. The decoder is a combination of the original decoder from StyleTTS and either iSTFTNet [45] or HifiGAN [28], with the AdaIN [44] module appended after each activation function. The duration and prosody predictors follow the same architecture as in [6], albeit with alterations to the output of the duration predictor changing from $1 \times N$ to $L \times N$ for probability output modeling. Thus, this section focuses primarily on providing comprehensive outlines for the style diffusion denoiser and the discriminative head of the SLM discriminators. Readers seeking detailed specifications for other modules can refer to the aforementioned references.

The **Denoiser** (Table 8) is designed to process an input style vector $s$, the noise level $\sigma$, the phoneme embeddings from the prosodic text encoder $h_{\text{bert}}$ and, when applicable, a speaker embedding $c$ from the acoustic and prosodic encoders in the multi-speaker scenario. Here, $c$ is also considered a part of the input and is introduced to the denoiser through adaptive layer normalization (AdaLN):

$$\text{AdaLN}(x, s) = L_\sigma(s)\frac{x - \mu(x)}{\sigma(x)} + L_\mu(s) \tag{27}$$

where $x$ is the feature maps from the output of the previous layer, $s$ is the style vector, $\mu(\cdot)$ and $\sigma(\cdot)$ denotes the layer mean and standard deviation, and $L_\sigma$ and $L_\mu$ are learned linear projections for computing the adaptive gain and bias using the style vector $s$.

Table 8: Denoiser architecture. $N$ represents the input phoneme length of the mel-spectrogram, $\sigma$ is the noise level, $h_{\text{bert}}$ is the phoneme embeddings, and $c$ is the speaker embedding. The size of $s$ and $c$ is $256 \times 1$, that of $h_{\text{bert}}$ is $768 \times N$, and $\sigma$ is $1 \times 1$. Group(32) is the group normalization with a group size of 32.

| Submodule | Input | Layer | Norm | Output Shape |
|---|---|---|---|---|
| Embedding | $\sigma$ | Sinusoidal Embedding | - | $256 \times 1$ |
| | $c$ | Addition | - | $256 \times 1$ |
| | – | Repeat for $N$ times | - | $256 \times N$ |
| | – | Linear $256 \times 1024$ | - | $1024 \times N$ |
| | – | Output Embedding $k$ | - | $1024 \times N$ |
| Input | $s$ | Repeat for $N$ Times | - | $256 \times N$ |
| | $h_{\text{bert}}$ | Concat | Group(32) | $1024 \times N$ |
| | – | Conv $1024 \times 1024$ | – | $1024 \times N$ |
| Transformer Block ($\times 3$) | $k$ | Addition | - | $1024 \times N$ |
| | $c$ | 8-Head Self-Attention (64 head features) | AdaLN $(\cdot, c)$ | $1024 \times N$ |
| | – | Linear $1024 \times 2048$ | – | $2048 \times N$ |
| | – | GELU | – | $2048 \times N$ |
| | – | Linear $2048 \times 1024$ | – | $1024 \times N$ |
| Output | – | Adaptive Average Pool | - | $1024 \times 1$ |
| | – | Linear $1024 \times 256$ | – | $256 \times 1$ |

The **Discriminative Head** (Table 9) is comprised of a 3-layer convolutional neural network, concluding with a linear projection.

Table 9: Discriminative head architecture. $T$ represents the number of frames (length) of the output feature $h_{\text{slm}}$ from WavLM.

| Layer | Norm | Output Shape |
|---|---|---|
| Input $h_{\text{slm}}$ | - | $13 \times 768 \times T$ |
| Reshape | - | $9984 \times T$ |
| Linear $9984 \times 256$ | - | $256 \times T$ |
| Conv $256 \times 256$ | - | $256 \times T$ |
| Leaky ReLU (0.2) | - | $256 \times T$ |
| Conv $256 \times 512$ | - | $512 \times T$ |
| Leaky ReLU (0.2) | - | $512 \times T$ |
| Conv $512 \times 512$ | - | $512 \times T$ |
| Leaky ReLU (0.2) | - | $512 \times T$ |
| Conv $512 \times 1$ | - | $1 \times T$ |

---

[15]Available at `https://github.com/yl4579/PL-BERT`

# Appendix G  Detailed Training Objectives

In this section, we provide detailed training objectives for both acoustic modules pre-training and joint training. We first pre-train the acoustic modules for accelerated training, and we then proceed with joint training with the pitch extractor fixed.

## G.1  Acoustic module pre-training

**Mel-spectrogram reconstruction.** The decoder is trained on waveform $\boldsymbol{y}$, its corresponding mel-spectrogram $\boldsymbol{x}$, and the text $\boldsymbol{t}$, using $L_1$ reconstruction loss as

$$\mathcal{L}_{\text{mel}} = \mathbb{E}_{\boldsymbol{x},\boldsymbol{t}} \left[ \left\| \boldsymbol{x} - M \left( G \left( \boldsymbol{h}_{\text{text}} \cdot \boldsymbol{a}_{\text{algn}}, \boldsymbol{s}_a, p_{\boldsymbol{x}}, n_{\boldsymbol{x}} \right) \right) \right\|_1 \right]. \tag{28}$$

Here, $\boldsymbol{h}_{\text{text}} = T(\boldsymbol{t})$ is the encoded phoneme representation, and the attention alignment is denoted by $\boldsymbol{a}_{\text{algn}} = A(\boldsymbol{x}, \boldsymbol{t})$. The acoustic style vector of $\boldsymbol{x}$ is represented by $s_a = E_a(\boldsymbol{x})$, $p_{\boldsymbol{x}}$ symbolizes the pitch F0 and $n_{\boldsymbol{x}}$ indicates energy of $\boldsymbol{x}$, and $M(\cdot)$ represents mel-spectrogram transformation. Following [6], half of the time, raw attention output from $A$ is used as alignment, allowing backpropagation through the text aligner. For another 50% of the time, a monotonic version of $\boldsymbol{a}_{\text{algn}}$ is utilized via dynamic programming algorithms (see Appendix A in [6]).

**TMA objectives.** We follow [6] and use the original sequence-to-sequence ASR loss function $\mathcal{L}_{\text{s2s}}$ to fine-tune the pre-trained text aligner, preserving the attention alignment during end-to-end training:

$$\mathcal{L}_{\text{s2s}} = \mathbb{E}_{\boldsymbol{x},\boldsymbol{t}} \left[ \sum_{i=1}^{N} \mathbf{CE}(\boldsymbol{t}_i, \hat{\boldsymbol{t}}_i) \right], \tag{29}$$

where $N$ is the number of phonemes in $\boldsymbol{t}$, $\boldsymbol{t}_i$ is the $i$-th phoneme token of $\boldsymbol{t}$, $\hat{\boldsymbol{t}}_i$ is the $i$-th predicted phoneme token, and $\mathbf{CE}(\cdot)$ denotes the cross-entropy loss function.

Additionally, we apply the monotonic loss $\mathcal{L}_{\text{mono}}$ to ensure that soft attention approximates its non-differentiable monotonic version:

$$\mathcal{L}_{\text{mono}} = \mathbb{E}_{\boldsymbol{x},\boldsymbol{t}} \left[ \left\| \boldsymbol{a}_{\text{algn}} - \boldsymbol{a}_{\text{hard}} \right\|_1 \right], \tag{30}$$

where $\boldsymbol{a}_{\text{hard}}$ is the monotonic version of $\boldsymbol{a}_{\text{algn}}$ obtained through dynamic programming algorithms (see Appendix A in [6] for more details).

**Adversarial objectives.** Two adversarial loss functions, originally used in HifiGAN [30], are employed to enhance the sound quality of the reconstructed waveforms: the LSGAN loss function $\mathcal{L}_{\text{adv}}$ for adversarial training and the feature-matching loss $\mathcal{L}_{\text{fm}}$.

$$\mathcal{L}_{\text{adv}}(G; D) = \mathbb{E}_{\boldsymbol{t},\boldsymbol{x}} \left[ \left( D \left( \left( G \left( \boldsymbol{h}_{\text{text}} \cdot \boldsymbol{a}_{\text{algn}}, \boldsymbol{s}_a, p_{\boldsymbol{x}}, n_{\boldsymbol{x}} \right) \right) \right) - 1 \right)^2 \right], \tag{31}$$

$$\mathcal{L}_{\text{adv}}(D; G) = \mathbb{E}_{\boldsymbol{t},\boldsymbol{x}} \left[ \left( D \left( \left( G \left( \boldsymbol{h}_{\text{text}} \cdot \boldsymbol{a}_{\text{algn}}, \boldsymbol{s}_a, p_{\boldsymbol{x}}, n_{\boldsymbol{x}} \right) \right) \right) \right)^2 \right] + \\ \mathbb{E}_{\boldsymbol{y}} \left[ \left( D(\boldsymbol{y}) - 1 \right)^2 \right], \tag{32}$$

$$\mathcal{L}_{\text{fm}} = \mathbb{E}_{\boldsymbol{y},\boldsymbol{t},\boldsymbol{x}} \left[ \sum_{l=1}^{\Lambda} \frac{1}{N_l} \left\| D^l(\boldsymbol{y}) - D^l \left( G \left( \boldsymbol{h}_{\text{text}} \cdot \boldsymbol{a}_{\text{algn}}, \boldsymbol{s}_a, p_{\boldsymbol{x}}, n_{\boldsymbol{x}} \right) \right) \right\|_1 \right], \tag{33}$$

where $D$ represents both MPD and MRD, $\Lambda$ is the total number of layers in $D$, and $D^l$ denotes the output feature map of $l$-th layer with $N_l$ number of features.

In addition, we included the truncated pointwise relativistic loss function [49] to further improve the sound quality:

$$\mathcal{L}_{\text{rel}}(G; D) = \mathbb{E}_{\{\hat{\boldsymbol{y}},\boldsymbol{y} | D(\hat{\boldsymbol{y}}) \leq D(\boldsymbol{y}) + m_{GD}\}} \left[ \tau - \text{ReLU} \left( \tau - \left( D\left( \hat{\boldsymbol{y}} \right) - D(\boldsymbol{y}) - m_{GD} \right)^2 \right) \right], \tag{34}$$

$$\mathcal{L}_{\text{rel}}(D; G) = \mathbb{E}_{\{\hat{\boldsymbol{y}}, \boldsymbol{y} | D(\boldsymbol{y}) \leq D(\hat{\boldsymbol{y}}) + m_{DG}\}} \left[ \tau - \text{ReLU}\left( \tau - (D(\boldsymbol{y}) - D(\hat{\boldsymbol{y}}) - m_{DG})^2 \right) \right], \quad (35)$$

where $\hat{\boldsymbol{y}} = G(\boldsymbol{h}_{\text{text}} \cdot \boldsymbol{a}_{\text{algn}}, \boldsymbol{s}_a, p_{\boldsymbol{y}}, n_{\boldsymbol{y}})$ is the generated sample, $\text{ReLU}(\cdot)$ is the rectified linear unit function, $\tau$ is the truncation factor that is set to be 0.04 per [49], $m_{GD}$ and $m_{DG}$ are margin parameters defined by the median of the score difference:

$$m_{GD} = \mathbb{M}_{\boldsymbol{y}, \hat{\boldsymbol{y}}} \left[ D(\hat{\boldsymbol{y}}) - D(\boldsymbol{y}) \right], \quad (36)$$

$$m_{DG} = \mathbb{M}_{\boldsymbol{y}, \hat{\boldsymbol{y}}} \left[ D(\boldsymbol{y}) - D(\hat{\boldsymbol{y}}) \right], \quad (37)$$

where $\mathbb{M}[\cdot]$ is the median operation.

**Acoustic module pre-training full objectives.** Our full objective functions in the acoustic modules pre-training can be summarized as follows with hyperparameters $\lambda_{\text{s2s}}$ and $\lambda_{\text{mono}}$:

$$\min_{G, A, E_a, F, T} \mathcal{L}_{\text{mel}} + \lambda_{\text{s2s}} \mathcal{L}_{\text{s2s}} + \lambda_{\text{mono}} \mathcal{L}_{\text{mono}} \qquad \min_{D} \mathcal{L}_{\text{adv}}(D; G)$$
$$\qquad (38) \qquad \qquad \qquad (39)$$
$$+ \mathcal{L}_{\text{adv}}(G; D) + \mathcal{L}_{\text{rel}}(G; D) + \mathcal{L}_{\text{fm}} \qquad \qquad + \mathcal{L}_{\text{rel}}(D; G)$$

Following [6], we set $\lambda_{\text{s2s}} = 0.2$ and $\lambda_{\text{mono}} = 5$.

## G.2 Joint training

**Duration prediction.** We use the following cross-entropy loss to train the duration predictor:

$$\mathcal{L}_{\text{ce}} = \mathbb{E}_{\boldsymbol{d}_{\text{gt}}} \left[ \sum_{i=1}^{N} \sum_{k=1}^{L} \mathbf{CE}\left( q[k, i], \mathbb{I}\left( \boldsymbol{d}_{\text{gt}}[i] \geq k \right) \right) \right], \quad (40)$$

where $\mathbb{I}(\cdot)$ is the indicator function, $q = S(\boldsymbol{h}_{\text{bert}}, \boldsymbol{s}_p)$ is the output of the predictor under the prosodic style vector $\boldsymbol{s}_p = E_p(\boldsymbol{x})$ and prosodic text embeddings $\boldsymbol{h}_{\text{bert}} = B(\boldsymbol{t})$, $N$ is the number of phonemes in $\boldsymbol{t}$, and $\boldsymbol{d}_{\text{gt}}$ is the ground truth duration obtained by summing $\boldsymbol{a}_{\text{algn}}$ along the mel frame axis.

Additionally, we employ the $L$-1 loss to make sure the approximated duration is also optimized:

$$\mathcal{L}_{\text{dur}} = \mathbb{E}_{\boldsymbol{d}_{\text{gt}}} \left[ \|\boldsymbol{d}_{\text{gt}} - \boldsymbol{d}_{\text{pred}}\|_1 \right], \quad (41)$$

where $\boldsymbol{d}_{\text{pred}}[i] = \sum_{k=1}^{L} q[k, i]$ is the approximated predicted duration of the $i^{\text{th}}$ phoneme.

**Prosody prediction.** We use $\mathcal{L}_{f_0}$ and $\mathcal{L}_n$, which are F0 and energy reconstruction loss, respectively:

$$\mathcal{L}_{f0} = \mathbb{E}_{\boldsymbol{x}} \left[ \|p_{\boldsymbol{x}} - \hat{p}_{\boldsymbol{x}}\|_1 \right] \quad (42)$$

$$\mathcal{L}_n = \mathbb{E}_{\boldsymbol{x}} \left[ \|n_{\boldsymbol{x}} - \hat{n}_{\boldsymbol{x}}\|_1 \right] \quad (43)$$

where $\hat{p}_{\boldsymbol{x}}, \hat{n}_{\boldsymbol{x}} = P(\boldsymbol{h}_{\text{bert}}, \boldsymbol{s}_p)$ are the predicted pitch and energy of $\boldsymbol{x}$.

**Mel-spectrogram reconstruction.** During joint training, we modify $\mathcal{L}_{\text{mel}}$ as follows:

$$\mathcal{L}_{\text{mel}} = \mathbb{E}_{\boldsymbol{x}, \boldsymbol{t}} \left[ \|\boldsymbol{x} - M(G(\boldsymbol{h}_{\text{text}} \cdot \boldsymbol{a}_{\text{algn}}, \boldsymbol{s}_a, \hat{p}_{\boldsymbol{x}}, \hat{n}_{\boldsymbol{x}}))\|_1 \right]. \quad (44)$$

Now we use the predicted pitch $\hat{p}_{\boldsymbol{x}}$ and energy $\hat{n}_{\boldsymbol{x}}$ to reconstruct the speech. We also make similar changes to $\mathcal{L}_{\text{adv}}$, $\mathcal{L}_{\text{fm}}$, and $\mathcal{L}_{\text{rel}}$ during joint training.

**SLM adversarial objective.** The SLM discriminator can easily overpower the generator and disable the gradient flow to the generator. To facilitate learning when the discriminator overpowers the generator, we use the LSGAN loss [48] instead of the min-max loss in eq. 5:

$$\mathcal{L}_{\text{slm}}(G; D) = \mathbb{E}_{\boldsymbol{t}, \boldsymbol{x}} \left[ \left( D_{SLM}\left( (G(\boldsymbol{h}_{\text{text}} \cdot \boldsymbol{a}_{\text{pred}}, \hat{\boldsymbol{s}}_a, p_{\hat{\boldsymbol{x}}}, n_{\hat{\boldsymbol{x}}})) \right) - 1 \right)^2 \right], \quad (45)$$

$$\mathcal{L}_{\text{slm}}(D;G) = \mathbb{E}_{\boldsymbol{t},\boldsymbol{x}}\left[\left(D_{SLM}\left(\left(G\left(\boldsymbol{h}_{\text{text}}\cdot\boldsymbol{a}_{\text{pred}}, \hat{\boldsymbol{s}}_a, p_{\hat{\boldsymbol{x}}}, n_{\hat{\boldsymbol{x}}}\right)\right)\right)\right)^2\right] +$$
$$\mathbb{E}_{\boldsymbol{y}}\left[\left(D_{SLM}(\boldsymbol{y}) - 1\right)^2\right], \tag{46}$$

where $\hat{s}_a$ is the acoustic style sampled from style diffusion, $\boldsymbol{a}_{\text{pred}}$ is the predicted alignment obtained through the differentiable duration upsampler from the predicted duration $\boldsymbol{d}_{\text{pred}}$, and $p_{\hat{\boldsymbol{x}}}$ and $n_{\hat{\boldsymbol{x}}}$ are the predicted pitch and energy with the sampled prosodic style $\hat{s}_p$.

**Joint training full objectives.** Our full objective functions in joint training can be summarized as follows with hyperparameters $\lambda_{\text{dur}}, \lambda_{\text{ce}}, \lambda_{f0}, \lambda_n, \lambda_{\text{s2s}}$, and $\lambda_{\text{mono}}$:

$$\min_{G,A,E_a,E_p,T,B,V,S,P} \mathcal{L}_{\text{mel}} + \lambda_{\text{ce}}\mathcal{L}_{\text{ce}} + \lambda_{\text{dur}}\mathcal{L}_{\text{dur}} + \lambda_{f0}\mathcal{L}_{f0} + \lambda_n\mathcal{L}_n$$
$$+ \lambda_{\text{s2s}}\mathcal{L}_{\text{s2s}} + \lambda_{\text{mono}}\mathcal{L}_{\text{mono}} + \mathcal{L}_{\text{adv}}(G;D) \tag{47}$$
$$+ \mathcal{L}_{\text{rel}}(G;D) + \mathcal{L}_{\text{fm}} + \mathcal{L}_{slm}(G;D) + \mathcal{L}_{\text{edm}},$$

where $\mathcal{L}_{\text{edm}}$ is defined in eq. 3.

The discriminator loss is given by:

$$\min_{D,C} \mathcal{L}_{\text{adv}}(D;G) + \mathcal{L}_{\text{rel}}(D;G) + \mathcal{L}_{slm}(D;G) \tag{48}$$

Following [6], we set $\lambda_{\text{s2s}} = 0.2, \lambda_{\text{mono}} = 5, \lambda_{dur} = 1, \lambda_{f0} = 0.1, \lambda_n = 1$ and $\lambda_{ce} = 1$.

