# OpenReview forum: "StyleTTS 2: Towards Human-Level Text-to-Speech through Style Diffusion and Adversarial Training with Large Speech Language Models"
_NeurIPS.cc/2023/Conference — NeurIPS 2023 poster_

### Official Review · Reviewer_mqLA · 2023-07-02

**Soundness:** 4 excellent
**Presentation:** 4 excellent
**Contribution:** 4 excellent
**Rating:** 6
**Confidence:** 5

**Summary:**

This work presents a GAN-based TTS model, as an improvement on its predecessor StyleTTS. Notably, it includes a diffusion-based style encoder, which is one major difference compared to its predecessor. In addition, the proposed model directly produce waveform, in contrast to spectrogram in its predecessor.



**Strengths:**

1) The proposed method is sound;
2) The audio samples sound great with clear improvements over multiple baselines
3) Abundant evaluations are conducted with persuasive results, including ablation studies

**Weaknesses:**

1) The proposed model is quite complicated
2) A few technical questions, see below

**Questions:**

1) "[StyleTTS 2] As the first model to achieve human-level performance on both single and multispeaker datasets" (line 47) -- Aren't PnG BERT and VITS achieved so as well, as acknowledged in Sec 2? If there is a difference on the scope of the claim, it's important to make it clear here. Also note that PnG BERT reported both MOS and CMOS (line 77).

2) Sec 3.2.1 claims "E2E training optimizes all TTS system components for inference without relying on any fixed components", but still "before jointly optimizing all components, we first pre-train the acoustic modules along with the pitch extractor and text aligner" -- is this still end-to-end training?

3) Sec 5.2: It's not clear how the evaluation is done. Are the styles analyzed the prosodic style, the acoustic style, or both? Are the emotion labels based on text, or audio (which can diverge)? what's the emotion of the reference audio? what about other factors as "style", beside emotion? Figure 2(c) -- "loose clusters" -- these are highly overlap to be considered as (even loose) "clusters"

4) Table 4: "real-time factor (RTF) in second" -- RTF is agnostic to the time unit

5) "SLM adversarial training" on "OOD texts" -- how does this work together with the regular training? There would be two types of training data, one is text-speech pairs, one is text-only -- how are they coordinated in an end-to-end manner? Can the model benefit from using model text-only data?

6) Evaluation on OOD text -- given the proposed model is trained on the OOD text (with SLM adversarial training), is it still a fair comparison to baseline models? importantly, are the eval text seen during SLM adversarial training?

7) Figure 1(b): where does $s_a$ come from during inference?

**Limitations:**

Any plan on releasing the source code and the trained checkpoints, especially given the claim on "StyleTTS 2 sets a new benchmark for TTS synthesis"?

---

> ### Author Rebuttal · Authors · 2023-08-06
>
> We are grateful for your positive remarks and appreciation of our work, and we appreciate your helpful feedback for clarification. Here are our responses to your insightful questions and concerns.
>
> **1. Claim of Human-Level Performance on Single and Multispeaker Datasets:**
>
> You are correct in pointing out that the achievements of PnG BERT and VITS are also human-level. However, VITS shows differences from ground truth in CMOS in the Appendix, indicating possible room for improvement. In addition, PnG BERT conducted the evaluation on a proprietary dataset rather than an open-access dataset. We appreciate your attention to detail, and we intend to revise the claim to specify that StyleTTS 2 is the first human-level model on publicly available single and multispeaker benchmark datasets. This revision will help in differentiating our contribution from previous works.
>
> **2. End-to-End Training:**
>
> The pre-training is only to speed up the training procedure because we found that having a well-trained acoustic model, particularly the style encoder in the first place can help the prosodic predictor converge faster. However, this is not absolutely necessary, and the model still converges when starting directly from the joint training without pre-training. Moreover, even with pre-training, all components are eventually trained together during joint training, so it is still considered end-to-end. We will make this point more clear in the revised version, by stating that this pre-training is not necessary for convergence.
>
> **3. Evaluation and Style Analysis:**
>
> The styles analyzed are concatenated acoustic and prosodic styles, and the emotion labels are based on texts generated by GPT-4. There is no reference for the single-speaker case shown in Figure 2(a), and the reference for the multispeaker case is neutral emotion (randomly picked from the testing set). This analysis aims to show that our model can sample different styles based on the emotions of the texts.
>
> There are definitely other aspects of styles that cannot be reflected in the text, though they can be reflected in the reference audio. Previous work (StyleTTS) has already shown that the framework can reproduce many aspects from the reference audio, so this work focuses more on the newly invented style diffusion. We have also provided samples on our demo page to show that our model can reflect several aspects of styles in the reference audio, including the acoustic environment and the speaker’s emotions.
>
> **4. Real-Time Factor (RTF):**
>
> We acknowledge the oversight in describing RTF, and we will correct this mistake in the camera-ready version.
>
> **5. SLM Adversarial Training on OOD Texts:**
>
> The training procedure in Figures 1(a) and 1(b) is indeed separate, as you have correctly noticed that their inputs are different, but the gradients are combined for each batch to update the parameters in the end. We will revise the caption under the figure to clarify this process and emphasize that adversarial training in Figure 1(b) is not together with joint training in Figure 1(a) because they have different inputs.
>
> **6. Evaluation on OOD Text:**
>
> The evaluation data was not seen during SLM adversarial training. In particular, the adversarial training uses texts from *train-clean-460* subset in the LibriTTS dataset, while in our evaluation we use LJSpeech, VCTK, and the *test-clean* subset of LibriTTS, so none of them are in the OOD texts used for training. We will make sure this is further emphasized in Section 4 Model Training.
>
> **7. Origin of $s_a$ in Figure 1b:**
>
> We apologize for the confusion in Figure 1b. The style diffusion model samples both $s_a$ and $s_p$, not just $s_p$. We will revise the figure and make the border around $s_a$ and $s_p$ darker in color to avoid any confusion.
>
> ***
> Lastly, we acknowledge that our method contains multiple components, but we hope that our paper has described each component sufficiently clearly. In addition, we do plan to make the source code publicly available to make sure all components in our method are completely transparent. We will add a link to the source code and checkpoints in the camera-ready version.

---

### Official Review · Reviewer_4B59 · 2023-07-05

**Soundness:** 4 excellent
**Presentation:** 3 good
**Contribution:** 4 excellent
**Rating:** 7
**Confidence:** 4

**Summary:**

In this paper, the paper proposes StyleTTS 2 for human-level text-to-speech, which is an end-to-end system with two-stage training (pretraining some modules and finetuning the whole network). Two main modules proposed in paper is style diffusion module to model the prosodic style information and WavLM adversarial module to improve the quality of synthesized waveforms. Moreover, the paper proposes an interesting module on duration prediction, which is differentiable and stable.

**Strengths:**

1. The paper proposes several interesting modules, including new differentiable duration modeling method, new adversarial module with WavLM and style diffusion to model the prosodic style information, which can benefit the future research in TTS.

2. The experiments verify the effectiveness and efficiency of proposed method by comparing MOS and RTF with different baseline methods.

3. The ablation study shows the effectiveness of proposed methods.

**Weaknesses:**

I do not see obvious weakness.

**Questions:**

1. For the multi-speaker setting or the zero-shot setting, how is the inference details. I think it will be different with Figure 1b since reference audio is used in inference?

2. Will the code be open-sourced to help the future research in TTS?

---

> ### Author Rebuttal · Authors · 2023-08-06
>
> We appreciate your positive assessment of our work and constructive feedback on our paper. Here are the responses to your questions.
>
> 1. Figure 1(b) represents the single speaker case only. In the multispeaker scenario, our model first encodes the speaker embedding (anchor style) and then samples a style using this anchor. This process is not well-articulated in the current version of the manuscript. We will revise the caption of Figure 1(b) to illustrate this process more clearly in the camera-ready version.
>
> 2. We will make the code publicly available to further the research in this field. We will include a link to the code in the camera-ready version of the paper.

---

### Official Review · Reviewer_ogQb · 2023-07-05

**Soundness:** 3 good
**Presentation:** 3 good
**Contribution:** 4 excellent
**Rating:** 7
**Confidence:** 4

**Summary:**

The paper presents Style TTS 2, a new TTS model with three main innovations 1) diffusion of a latent variable to capture style (everything in the speech signal that’s not enumerated by a phone sequence t and speaker variable c), 2) the use of a pretrained speech language model (SLM) as a discriminator and 3) a novel formulation of a differentiable alignment function.  The authors also enable a new joint training curriculum.   These three innovations result in quality that surpasses existing available TTS approaches including VITS, VALL-E, Natural Speech and YouTTS.


**Strengths:**

The paper is well written, and, for the most part, easy to follow.   Each of the novel components are clearly described, well motivated and contextualized with other comparable approaches. Also, the quality of the samples is quite good.

**Weaknesses:**

Given that zero-shot speaker adaptation is extremely ripe for misuse as Societal Impact and Potential Harmful Consequences
per https://neurips.cc/public/EthicsGuidelines, I would think the risk and mitigation section of this presentation should be more than 2 sentences.

Section 3.2.1 using less resources is highlighted as a contribution of Style TTS 2 – does this include the “phoneme-level BERT pretrained on extensive corpora of Wikipedia articles”?  Similar question concerning Section 3.2.3 and the use of WavLM pre-trained on 94k hours of data?  It seems like this assessment is limited to the resources required for the model itself, but does not include components like the discriminator and text embedding.

Some areas to tighten the presentation and contribution in the upfront material of the paper.

Introduction: the first paragraph establishes the areas of improvement for TTS as diverse, expressive speech, robustness to OOD text, and requirements of massive datasets for performing zero-shot TTS.   In the last paragraph of the introduction the contributions are oriented about quality, and novel use of SLMs. However, these are inconsistent with the areas for improvement enumerated by the authors.  It might be clearer to be specific about which open questions for in the first paragraph are going to be addressed in some form by the proposed technique.

The limitations of StyleTTS are again not the same as those established in the introduction. Namely they are 1) a two-stage training process, 2) limited expressiveness and 3) reliance on reference speech hindering realtime application.  StyleTTS 2 addresses these three limitations, unifying the training process, and reducing reliance of reference speech while improving OOD performance and expressiveness.  While this is compact and consistent as an improvement ot StyleTTS, it does not address the broad limitations of TTS established by the authors.

Section 3.2 and 3.2.1 one of the advantages of Style TTS 2 is described as an E2E training process that jointly optimizes all components.  This is presented in contrast to STyle TTS which required a two-stage training process (Section 3.1).  However, later in Section 3.2.1 (line 142 and Figure 1a) Style TTS2 uses a two-stage training pre-training and joint training process.  If single stage training, or E2E training is meant to be an advantage and contribution of Style TTS 2, the distinction between its two-stage training and the two-stage training used by the original Style TTS needs to be more clearly defined.


**Questions:**

What does it mean for synthetic samples to “surpass” human recordings? Not in the context of how this happens as a score, but what does it say about the evaluation itself if it’s possible for an artificial sample to be more “natural” than a human derived sample which is “natural” with metaphysical certitude.  I appreciate the argument about unnatural segmentation of the audio in LJSpeech, but if this is the source of the “unnaturalness” in the test data, it’s maybe not worth making too much of a claim about its value as a quantitative evaluation.

On line 151, the augmented style vector is defined as s = [s_p,s_d], but s_d is not previously defined.  Should this be s_a?

Section 3.2.3 - the generator loss being independent of ground truth is true of any GAN training, no? Is there something specific to the use of an SLM discriminator that enables this better than prior work training TTS with GAN objectives?

Section 5.1 “setting a new standard for this dataset”.  What is the referent of “this dataset”? Is it the evaluation set of 80 text samples from LIbriTTS and 40 Librivox utterances spoken by the speaker of LJSpeech described in Section 4?


**Limitations:**

Very briefly in two sentences at the end of the conclusion. (My rating below is contingent on the outcome of an ethics review to assess if the limitations and discussion of ethics concerns are sufficient to comply with https://neurips.cc/public/EthicsGuidelines)

---

> ### Author Rebuttal · Authors · 2023-08-06
>
> Thank you for your thorough review and insightful comments on our work. We appreciate your observations and feedback that have helped us identify areas for clarification and improvement, and here are our responses.
>
> - **Ethical Concerns on Misuse**
>
> We are grateful for your attention to the potential ethical problems in our work. We acknowledge that having only two sentences is insufficient for addressing ethical issues of potential misuse. We will expand the discussion in the conclusion section by adding the following paragraph:
>
> > We acknowledge that zero-shot speaker adaptation has the potential for misuse and deception by mimicking the voices of individuals as a potential source of misinformation or disinformation. This could lead to harmful, deceptive interactions such as theft, fraud, harassment, or impersonations of public figures that may influence political processes or trust in institutions. In order to manage the potential for harm, we will require users of our model to adhere to a code of conduct that will be clearly displayed as conditions for using the publicly available code and models. In particular, we will require users to inform those listening to samples synthesized by StyleTTS 2 that they are listening to synthesized speech or to obtain informed consent regarding the use of samples synthesized by StyleTTS 2 in experiments. Users will also be required to use reference speakers who have given consent to have their voice adapted, either directly or by license. Finally, we will make the source code publicly available for further research in speaker fraud and impersonation detection.
>
> - **Resources for Zero-Shot Speaker Adaptation Task**
>
> We acknowledge your concern regarding the resources utilized in our zero-shot speaker adaptation task. Our intention was to emphasize the reduction in the need for speech data in TTS model training rather than the entire training pipeline. Moreover, since pre-trained models like PL-BERT and WavLM are already publicly available, we do not consider the training data used for these models a part of required training resources. Compared to models like Vall-E that trains on 60k hours of data, our approach can be seen as a resource-efficient method, as we are using 250 times less speech data than Vall-E while achieving similar or better performance. We will revise our paper to make this point more straightforward.
>
> - **Addressing Limitations in Previous Works**
>
> Thank you for pointing out this inconsistency in presenting the areas for improvement and our model’s contributions in the introduction. Our model indeed addresses the limitations mentioned in the introduction, but it would be easier to follow if we were more consistent in our wording so that it was clear that our model contributed solutions to these limitations. Using style diffusion, our model can synthesize diverse samples, outperforming models like VITS, as shown in Section 5.2 and Appendix A.2.1. The adversarial training with SLM also improves the performance on out-of-distribution (OOD) texts. Furthermore, our model performs comparably to Vall-E, despite using significantly less speech data, making our model a resource-efficient alternative to models requiring a large amount of speech data like Vall-E. We appreciate your suggestion and we will highlight these contributions more in the conclusion section for consistency.
>
> - **Meaning of Outperforming Human Recordings**
>
> Thank you for bringing up the meaning of outperforming human data in the evaluation of naturalness. There are important considerations to make about the way in which speech segments were evaluated. First, the evaluated segments were originally spoken in a larger context. It is possible that a certain speech segment spoken in isolation sounds less natural than the same speech spoken in context.
> However, StyleTTS 2 synthesizes segments with a more limited context which may sound more natural in isolation than in a larger context. This would lead to higher ratings for StyleTTS 2 than for real speech, as the lack of larger context in evaluation would favor synthesized speech. Second, human speech naturally contains variability that is neither text nor context-dependent. This variability can lead to short segments of real speech sounding less natural than expected on average. On the other hand, synthesized speech lacks this variability, producing “average” speech given the current text and context, which sounds more natural on average. We will include these discussions in the revised paper to guide future research in this area.
>
> - **Clarification on Two-Stage Training Process**
>
> We understand the confusion around our two-stage training process. Unlike the original StyleTTS, which fixed the acoustic modules during the second stage, StyleTTS 2 goes through joint training of both acoustic and predictor modules after pre-training. The initial pre-training is primarily aimed at accelerating the training process, not a strict necessity. We found that having a well-trained acoustic module first, particularly the acoustic style encoder as the starting point for the parameters of the prosodic encoder, promotes faster convergence of the prosody predictor. We will clarify this in the camera-ready version.
>
> - **Typos and Other Clarifications**
>
> We appreciate your attention to detail, and we will correct the notation mistake on line 151.
>
> We agree that the independence of the generator from the ground truth is common in GAN training. However, in our work, this independence allows us to train on OOD texts without specific speech ground truth, improving performance on OOD texts. We will emphasize more on this point in the revision.
>
> We also apologize for the ambiguity in Section 5.1. By "this dataset", we were referring to the in-distribution samples from the LJSpeech dataset, as our model has outperformed the previous state-of-the-art model NaturalSpeech on this dataset. We will make this more specific in the revised paper.

---

> > ### Comment · Reviewer_ogQb · 2023-08-18
> > **thank you**
> >
> > Thank you very much for answering my questions! I enjoyed reading the work and hope my comments helped to strengthen its presentation.
> >
> > I'll leave the assessment of the ethics comments to the ethics reviewers.

---

### Official Review · Reviewer_5CmP · 2023-07-06

**Soundness:** 3 good
**Presentation:** 3 good
**Contribution:** 3 good
**Rating:** 7
**Confidence:** 4

**Summary:**

The authors of the paper have proposed a high-quality TTS model StyleTTS 2. Unlike previous StyleTTS work, StyleTTS 2 has a DDPM-based style generator, speech language model discriminators, and a differentiable aligner. All of these design decisions help improve
the overall performance of the model. As the authors report, StyleTTS 2 sets new standards for TTS synthesis.

**Strengths:**

I appreciate the hard work of the authors on this paper. It was a well-written and enjoyable read.
It is a great challenge to further develop TTS models with high quality, as the current SOTA models are very close to human speeches.
StyleTTS 2 optimizes all the components in the joint training stage.
Evaluation is thorough enough to support the arguments with in-depth ablation studies.
I have carefully checked the audio samples, they are promising.

**Weaknesses:**

The training procedures are complex. It requires several training stages, such as pre-training the acoustic modules, retraining the speech language models, and joint training.
It is unclear whether the speech language models (SLMs) are trained on audio clips (rather than the whole-length audio). If so, how do you learn semantic information from short audio clips?
It is also unclear whether the MPD and MRD still exist in the jointly training stage. I found MPD and MRD missing in Figure 1 (b).
The overall novelty is limited, as many previous works integrate the diffusion modules and speech language model modules.

**Questions:**

Do MPD and MRD still exist in the jointly training stage?

---

> ### Author Rebuttal · Authors · 2023-08-06
>
> Thank you for your time and effort in reviewing our paper and your constructive questions. We appreciate your positive feedback on our work and your insightful comments. We would like to address your concerns and clarify certain aspects of our paper.
>
> - **Concern about Complicated Training Procedure**
>
> We acknowledge that the training procedure consists of more than one step. We hope that we can alleviate this issue by making the training code publicly available, ensuring that users can train our model easily. In addition, we would like to clarify that the SLMs are not re-trained. Instead, we appended a discriminator head on top of it, as mentioned in Section 3.2.3. We will ensure to highlight this point more prominently in the revised manuscript.
>
> - **Concern about Training SLM Discriminators on Audio Clips**
>
> We followed the methodology of the original WavLM paper for downstream tasks. For example, the original WavLM paper used the SLM representations on 3-second clips and fine-tuned for extra 2 epochs with 6-second clips for the speaker verification task. They also used a batch size of 200 seconds of audio for base model finetuning and 80 seconds of audio for large model finetuning for the speech recognition task. In our training process, we randomly cropped the audio into 3-6 seconds (thus 96 to 192 seconds per batch) to mimic how the WavLM model was used for downstream tasks, as we have described in Section 4.
>
> However, we acknowledge that having longer clips can help with the performance, but it is also more costly in terms of GPU RAM usage. We have found that 3-6 seconds of clips are sufficiently long to produce naturalistic speech. Since the semantic information the discriminator captures is primarily for understanding the paralinguistic information as described in Appendix D.1, we stick to this setting in our implementation. We will make this point more apparent in the revised paper to avoid confusion.
>
> - **Concern about the Absence of MPD and MRD in Figure 1(b)**
>
> Figure 1(b) is an illustration of adversarial training with SLM only. MPD and MRD are not present for adversarial training with SLM but are present during the joint training process, as shown in Figure 1(a). The training consists of both joint training and adversarial training with SLM, where the gradients from both processes are combined to update the parameters of trainable modules in each batch.
>
> We agree that the absence of these components in Figure 1(b) could be confusing. In the revised manuscript, we will amend the caption of Figure 1 to more clearly show that the actual training procedure is a combination of the processes illustrated in Figures 1(a) and 1(b).
>
> - **Concern about Novelty**
>
> We appreciate your feedback, and we apologize for not presenting the novelties of our work as clearly as possible. The main novelties that we want to present are the following: Unlike previous works that sample waveforms, melspectrograms, or hidden representations proportional to the duration of the speech using diffusion models, we are the first to diffuse a fixed-length vector for speech synthesis, which greatly improves the model efficiency while maintaining the benefits of diverse samples provided by diffusion models. Moreover, unlike previous works, we employ SLM as a discriminator, rather than a source of hidden representations from which speech is decoded or reconstructed. While diffusion models and SLMs have indeed been proposed in previous works for speech synthesis, we use them in new ways that greatly improve the performance of the model. We will emphasize these points more in Section 2 of the final version of the manuscript.

---

> > ### Comment · Reviewer_5CmP · 2023-08-17
> > **Thanks for the comments**
> >
> > Thanks for your comments. I believe this paper merits acceptance as  it is a high-quality paper, with a significant contribution that would be of interest to the community.

---

### Official Review · Reviewer_JmdN · 2023-07-08

**Soundness:** 2 fair
**Presentation:** 1 poor
**Contribution:** 2 fair
**Rating:** 5
**Confidence:** 4

**Summary:**

The paper introduces StyleTTS 2, a text-to-speech (TTS) model that leverages latent diffusion and adversarial training with SLMs. The most noteworthy thing is that the paper employs large pre-trained speech language models (SLMs), such as WavLM, as discriminators in the adversarial training process. This novel approach, combined with differentiable duration modeling for end-to-end training, improves the naturalness of the synthesized speech.

**Strengths:**

 1. It utilizes adversarial training with large speech language models (SLMs) in conjunction with differentiable duration modeling for end-to-end training. This approach improves the naturalness of synthesized speech.

2. Text encoder in StyleTTS 2 is separated into an Acoustic Text Encoder and a Prosodic Text Encoder. The Prosodic Text Encoder is then used as a conditional input for the latent diffusion process. This design allows the Diffusion model to model the diverse prosody of different styles, and it can also be seen through the ablation study that latent diffusion is essential in this paper.



**Weaknesses:**

1. The article fails to provide a clear explanation of how the prosodic style encoder ensures the extraction of style from the Mel spectrogram rather than other elements. The specific mechanism behind this process is not adequately detailed in the paper. While the article briefly mentions certain style diffusion methods, they do not sufficiently explain how the prosodic style is accurately derived from the Mel spectrogram. Further exploration is required to devise more effective encoder architectures and training mechanisms to ensure the precise extraction of prosodic style information.

2. The writing style of the article lacks coherence, making it challenging to grasp the main ideas and problem-solving strategies. The organization and presentation of the paper could have been improved.

-----------
The author's response addressed some of my issues, so I raised my score.

**Questions:**

None

**Limitations:**

Yes

---

> ### Author Rebuttal · Authors · 2023-08-06
>
> We are grateful for your valuable time and effort in reviewing our paper. Your insights have helped us clarify certain points about our work. Here are our point-to-point responses.
>
> - **Concern about Zero-Shot TTS and Prosodic Styles**
>
> We acknowledge that this aspect may not have been clearly articulated, and we apologize for not clarifying that the inference pipeline in Figure 1(b) is for the single-speaker model only. For multispeaker models, including zero-shot speaker adaptation, the style diffusion model does take both acoustic and prosodic styles as input, as described in the last paragraph of Section 3.2.2. We will make it clear that Figure 1(b) represents the single-speaker case only and we will add an additional description for the multispeaker case under the figure in the camera-ready version. Moreover, we would like to emphasize that our primary contribution lies in presenting human-level TTS systems for seen speakers with style diffusion and SLM discriminators. Beyond this, we present the zero-shot feature as an additional capability, and we leave further improvements in zero-shot speaker adaptation for future works, as discussed in Section 6.
>
> - **Concern about Prosodic Style Encoder and Decoupling from Speaker Timbre**
>
> We regret that we have not made the function of our prosodic style encoder more clear. Its role is not to decouple from the speaker timbre but to capture additional stylistic information that the acoustic style encoder cannot. This relieves the "burden" of the acoustic style encoder so the information it extracts does not have to be used to both reconstruct the speech and predict the prosody (duration and F0) faithfully. As you have correctly pointed out,  the former needs to encode more of the timbre information, while the latter needs to extract more of the prosodic aspects. Hence, the prosodic style encoder is designed to complement, not decouple, the speaker's timbre by capturing additional stylistic information. We agree that this point can be made more clear and we will make the motivation of adding the prosodic style encoder more straightforward in the revision.
>
> In addition, since the F0 is predicted based on the style extracted from the reference audio, our model is able to produce speech with pitch ranges similar to the reference speech. This was already shown in the previous work StyleTTS, and we have presented several samples in Section 4 on our demo page to demonstrate this effect. We acknowledge that this could have been elaborated more clearly, and we will do so in the revision by emphasizing that our model maintains the capability of the original StyleTTS that can match the styles in the synthesized speech with those extracted from the reference audio such as the pitch range.
>
> - **Concern about Clarity and Understanding of Prosodic Style Encoder**
>
> We would like to clarify that the prosodic style encoders are trained specifically to reconstruct duration, F0, and energy, which we refer to as "prosody". It is worith noting that the prosodic style encoder is not the main focus of our paper. It is only to address the problem we found in the previous work (StyleTTS) that using a single style encoder to reconstruct both the speech and the prosodic information (duration, F0, and energy) is not sufficient. Hence, we simply duplicated the original style encoder to encode the acoustic and prosodic information separately. We appreciate the reviewer’s suggestion for further exploration of encoder architectures and training mechanisms, which we plan to do in future work. We will discuss this in Section 6 as a possible direction for future research in the revised version.
>
> - **Concern about the Readability of the Paper**
>
> We appreciate your feedback regarding the readability of our paper. We have made significant efforts to improve the readability and accessibility of the writing. Based on your feedback and that of the other reviewers, we have improved the clarity of several sections. We believe the overall coherence is greatly improved. We welcome further comments or suggestions on how we can improve the presentation of our work.

---

### Author Rebuttal · Authors · 2023-08-06

We thank all the reviewers for their time and effort in reviewing and improving our work. Here are some common concerns we would like to address:

***

**Source Code and Checkpoints**

We will make the trained checkpoints and source code for training and inference publicly available. We will add a link to the GitHub repository in the camera-ready version that contains the source code and checkpoints. To prevent potential misuse for deception, we will release the model under a code of conduct that the listeners of the samples should be informed that the speech they are listening to is synthesized by StyleTTS 2, and the reference speakers should have given consent either directly or by license to have their voice adapted by StyleTTS 2.

***

**Figure 1 Clarification**

We have noticed several confusions caused by unclarity in our presentation in Figure 1, and we will make the following changes:

- We will clarify that Figure 1 illustrates two separate training processes in Figure 1(a) and 1(b), with different inputs but combined gradients for each batch.

- We will make it clear that Figure 1(b) only describes the inference pipeline for single-speaker models, and we will add additional descriptions for multispeaker cases involving reference speaker embeddings.

- We will clarify that the joint training process described in Figure 1(a) is end-to-end with all components being trained together. We will emphasize that the pre-training described in Figure 1(a) is for faster convergence only, not an absolute necessity.

***

**Ethics Review Rebuttal**

We appreciate the ethics reviewer's careful evaluation of any ethical issues in our work. Here is our rebuttal to the ethics review and our plan to address these concerns.

**1. IRB Details**:

We thank the ethics reviewer for noting that we omitted details about the IRB approval of the protocol used to obtain speech evaluations from human participants. We will make revisions to include further details about the human study protocol.

Previous:

> “These evaluations were conducted by native English speakers from the U.S. on Amazon Mechanical Turk.”

Updated:

>“These evaluations were conducted by native English speakers from the U.S. on Amazon Mechanical Turk. All evaluators reported normal hearing and provided informed consent as monitored by the local institutional review board and in accordance with the ethical standards of the Declaration of Helsinki.”

We will further disclose the IRB protocol number in the camera-ready version as proof.

**2. Potential for Deception and Ways for Mitigation:**

We appreciate the importance of the potential for deception this model creates. We will address this issue more wholistically by noting further impacts of possible deception and our specific plans for mitigating possible negative consequences of the release of this model. Specifically, we will expand our discussion in Section 6 with the following paragraph:

>We acknowledge that zero-shot speaker adaptation has the potential for misuse and deception by mimicking the voices of individuals as a potential source of misinformation or disinformation. This could lead to harmful, deceptive interactions such as theft, fraud, harassment, or impersonations of public figures that may influence political processes or trust in institutions. In order to manage the potential for harm, we will require users of our model to adhere to a code of conduct that will be clearly displayed as conditions for using the publicly available code and models. In particular, we will require users to inform those listening to samples synthesized by StyleTTS 2 that they are listening to synthesized speech or to obtain informed consent regarding the use of samples synthesized by StyleTTS 2 in experiments. Users will also be required to use reference speakers who have given consent to have their voice adapted, either directly or by license. Finally, we will make the source code publicly available for further research in speaker fraud and impersonation detection.

**3. Meaning and Implications of Outperforming Human Data**

We thank the ethics reviewer for bringing up the meaning of outperforming human data on an evaluation of naturalness. There are important considerations to make about the way in which speech segments were evaluated. First, the evaluated segments were originally spoken in a larger context. It is possible that a certain speech segment spoken in isolation sounds less natural than the same speech spoken in context. However, StyleTTS 2 synthesizes segments with a more limited context which may sound more natural in isolation than in a larger context. This would lead to higher ratings for StyleTTS 2 than for real speech, as the lack of larger context in evaluation would favor synthesized speech. Second, human speech naturally contains variability that is neither text- nor context-dependent. This variability can lead to short segments of real speech sounding less natural than expected on average. On the other hand, synthesized speech lacks this variability, instead producing “average” speech given the current text and context, which sounds more natural on average.

Given that we can synthesize speech barely distinguishable from real speech, it raises the question of how real speech could be reasonably detected. Since we believe StyleTTS 2 outperforms human data mainly due to context and variability, we believe these characteristics could also be used for detecting synthesized speech. In particular, shorter segments may not be detected accurately, as they lack context and only represent a small amount of the possible variation in the speaker’s voice. On the other hand, longer segments may be detected more accurately, as real speech will be more influenced by long-term context and contain more “human” sources of variability, such as pauses and breaths. We will include these discussions in the revised paper to guide future research in this area.

---

### Decision · Program_Chairs · 2023-09-21

**Decision:**

Accept (poster)

**Comment:**

This paper introduces a high-quality TTS model StyleTTS 2, which has a DDPM-based style generator, speech language model discriminators, and a differentiable aligner. Reviewers are all positive about this paper. Some reviewers also have concerns about the complex training procedure of the model. Suggest the author validate all the components to verify whether they are unnecessarily complicated. Overall, it is a good paper, and I recommend to accept.